EMBO
Molecular Medicine

# MET inhibition overcomes radiation resistance of glioblastoma stem-like cells

Francesca De Bacco[1], Antonio D'Ambrosio[1,2], Elena Casanova[1], Francesca Orzan[1], Roberta Neggia[1], Raffaella Albano[1], Federica Verginelli[1], Manuela Cominelli[3], Pietro L Poliani[3], Paolo Luraghi[1], Gigliola Reato[1,2], Serena Pellegatta[4], Gaetano Finocchiaro[4], Timothy Perera[5], Elisabetta Garibaldi[6], Pietro Gabriele[6], Paolo M Comoglio[1,2,*] & Carla Boccaccio[1,2,**]

## Abstract

Glioblastoma (GBM) contains stem-like cells (GSCs) known to be resistant to ionizing radiation and thus responsible for therapeutic failure and rapidly lethal tumor recurrence. It is known that GSC radioresistance relies on efficient activation of the DNA damage response, but the mechanisms linking this response with the stem status are still unclear. Here, we show that the MET receptor kinase, a functional marker of GSCs, is specifically expressed in a subset of radioresistant GSCs and overexpressed in human GBM recurring after radiotherapy. We elucidate that MET promotes GSC radioresistance through a novel mechanism, relying on AKT activity and leading to (i) sustained activation of Aurora kinase A, ATM kinase, and the downstream effectors of DNA repair, and (ii) phosphorylation and cytoplasmic retention of p21, which is associated with anti-apoptotic functions. We show that MET pharmacological inhibition causes DNA damage accumulation in irradiated GSCs and their depletion *in vitro* and in GBMs generated by GSC xenotransplantation. Preclinical evidence is thus provided that MET inhibitors can radiosensitize tumors and convert GSC-positive selection, induced by radiotherapy, into GSC eradication.

**Keywords** glioblastoma; glioblastoma stem-like cells; MET oncogene; radiosensitization; radiotherapy

**Subject Categories** Cancer; Neuroscience

## Introduction

Treatment of GBM, a rapidly lethal primary brain tumor, has little advanced in the last decade. Although the genetic and molecular features of the disease have been elucidated, the high degree of inter- and intratumor heterogeneity restrains the use of targeted therapies (Brennan *et al*, 2013; Ene & Holland, 2015). As a result, GBM standard of care is still based on maximal but often incomplete surgical resection, combined with chemoradiotherapy (Stupp *et al*, 2009). The short-lived therapeutic response has been associated with the finding that ionizing radiations (IR) destroy the tumor bulk, but not GSCs, which drive tumor recurrence (Bao *et al*, 2006). The higher radioresistance of GSCs, as compared with non-stem cells, correlates with hyperactivation of the DNA damage response (DDR), and escape from apoptosis (Bao *et al*, 2006; Cheng *et al*, 2011). Moreover, it has been envisaged that radioresistance can rely on signaling from tyrosine kinase receptors, such as the frequently amplified EGFR (Squatrito & Holland, 2011), but the issue has not been addressed in GSCs.

Recently, we and others showed that MET, the HGF receptor, sustains radioresistance of conventional cell lines (Lal *et al*, 2005; De Bacco *et al*, 2011). We then showed that MET is expressed in ~40% of GSC lineages, derived as neurospheres (i.e. *in vitro* cultures enriched in stem and progenitor cells) from GBM patients (De Bacco *et al*, 2012). Interestingly, these "MET-pos-NS" usually lack EGFR amplification and display a mesenchymal or proneural transcriptional profile (De Bacco *et al*, 2012), according to Verhaak *et al* (2010). We also showed that, although clonal, MET-pos-NS contain cells expressing different levels of MET. The sorted MET^high and MET^neg subpopulations display opposite features, with MET^high retaining GSC properties such as (i) long-term self-propagating and multi-potential differentiation ability *in vitro*; (ii) tumorigenic property *in vivo*; and (iii) ability to reconstitute a mixed MET^high/MET^neg

1 Laboratory of Cancer Stem Cell Research, Candiolo Cancer Institute, FPO-IRCCS, Candiolo, Italy
2 Department of Oncology, University of Torino, Candiolo, Italy
3 Department of Molecular and Translational Medicine, Pathology Unit, University of Brescia, Brescia, Italy
4 Unit of Molecular Neuro-Oncology, Fondazione IRCCS Istituto Neurologico C. Besta, Milan, Italy
5 Octimet Oncology Ltd., Oxford, UK
6 Unit of Radiotherapy, Candiolo Cancer Institute, FPO-IRCCS, Candiolo, Italy
*Corresponding author. Tel: +39 011 9933 601; Fax: +39 011 9933 225; E-mail: direzione.scientifica@ircc.it
**Corresponding author. Tel: +39 011 9933 208; Fax: +39 011 9933 225; E-mail: carla.boccaccio@ircc.it

 

population *in vitro* and *in vivo*. In contrast, the MET[neg] subpopulation lacks the above properties and displays prodifferentiating features (De Bacco *et al*, 2012). Thus, we concluded that MET marks a cell subpopulation that regenerates self-renewing MET-positive cells, and a progeny losing GSC properties along with MET expression. Finally, we showed that MET is a functional marker, as HGF sustains GSC self-renewal (De Bacco *et al*, 2012). Accordingly, other studies showed that MET is a marker for prospective GSC isolation from patients (Joo *et al*, 2012) and sustains the GSC phenotype by upregulating reprogramming transcription factors (Li *et al*, 2011).

On these premises, we investigated whether MET is a functional marker of GSC radioresistance, and whether MET could be exploited as a therapeutic target to achieve radiosensitization of GBM by depletion of its GSC component.

# Results

## Genetic and phenotypic characterization of a large panel of neurospheres and correlations with radioresistance

We investigated whether patient-derived GSCs, propagated as NS, displayed greater radioresistance than non-stem/differentiated cells of the same lineage and whether radioresistance could be preferentially associated with specific genetic alterations or transcriptional profiles. A panel of 106 NS derived from primary GBM underwent analysis of the most frequent GBM genetic alterations such as EGFR amplification (EGFR[amp]), PTEN inactivation by deletion or mutation (PTEN[loss]), TP53 mutation, and NFKBIA deletion. NS were then classified as proneural, classical, or mesenchymal by transcriptional profiling, according to the Verhaak's signature (Verhaak *et al*, 2010). A panel of 20 NS equally distributed among the three subtypes was chosen (Appendix Tables S1 and S2). The response of these NS to therapeutic doses of IR was assessed and compared with that of astrocytes (SVG, NHA) and GBM cell lines (U251, SNB19), and the same NS cultured in prodifferentiating conditions (Galli *et al*, 2004; De Bacco *et al*, 2012). In dose–response and time–course viability experiments, all NS invariably displayed higher radioresistance than astrocytes or GBM cell lines, or the NS pseudodifferentiated progeny (Figs 1A and B and EV1A and Appendix Fig S1A). No significant viability change was observed among different NS (Fig 1A), nor it was associated with any of the genetic alterations tested or any transcriptional subtype (Fig EV1B and C).

Next, NS radioresistance was correlated with the ability to evoke DDR. Phosphorylation of histone H2AX (γH2AX), a key step in the nucleation of DNA repair complex at double-strand breaks (DSBs), was measured by flow cytometry (Olive, 2004). Early after irradiation, γH2AX levels were comparable in NS, astrocytes, GBM cell lines, and NS-derived pseudodifferentiated cells (~10-fold over basal levels). Within 6 h, γH2AX resolution occurred in NS, indicating DSB repair, while γH2AX persisted, or even increased in the NS-derived pseudodifferentiated cells, suggesting DSB persistence (Fig 1C and Appendix Fig S1B). Consistently, comet assays showed that, while a low level of DNA damage was detectable in NS 24 h after irradiation, heavy damage remained in astrocytes, GBM cell lines, and, mainly, in the NS pseudodifferentiated progeny (Fig 1D).

The greater ability of NS to repair DSBs, as compared with differentiated cells, was found to correlate with increased activation of two crucial DDR signal transducers: (i) ATM, the apical kinase unleashed by DNA damage sensors, responsible for H2AX phosphorylation (Shiloh & Ziv, 2013); and (ii) the ATM substrate Chk2, whose activation correlates with GSC radioresistance (Bao *et al*, 2006) (Fig 1E and Appendix Fig S1C). Accordingly, the DNA recombinase RAD51, upregulated at the DDR cascade end and critical for homologous recombination repair (van Gent *et al*, 2001), was significantly increased in NS, but not in their pseudodifferentiated progeny or astrocytes (protein: Fig 1E and Appendix Fig S1C; mRNA: Fig 1F). As a result, 24 h after irradiation, GBM cell lines or NS-derived pseudodifferentiated cells displayed cell cycle arrest and accumulation in G2-M phase, while NS resumed proliferation, as indicated by the increased number of S-phase cells (Appendix Fig S1D–F).

Collectively, these findings indicate that GSCs activate DDR more rapidly and effectively than cells endowed with differentiated features, resulting in efficient DNA repair, survival, and proliferation after irradiation.

## The neurosphere GSC subpopulation is selected by irradiation

By definition, NS are clones deriving from expansion of single self-renewing GSCs. However, each NS entails a certain degree of phenotypic heterogeneity, which arises within the progeny of the founder GSC, and implies loss of stem properties, including radioresistance. We thus investigated whether, within the whole heterogeneous NS, the GSC subpopulation was more radioresistant than its non-stem progeny and whether it was positively selected by IR. It is known that stem-like cells from GBM and other tumors can be identified by retention of PKH-26 dye (Pece *et al*, 2010; Richichi *et al*, 2013).

**Figure 1.  GSCs are radioresistant.**

A   Cell viability of NS, GBM, and astrocyte cell lines measured 72 h after irradiation (fold versus non-irradiated cells, ctrl). *: one-way ANOVA, $P = 0.0001$. $n = 2$.

B   Cell viability of BT308NS and their pseudodifferentiated progeny (BT308 diff) measured 24 h after irradiation (fold versus non-irradiated cells, ctrl). *: t-test, $P = 0.0006$. $n = 2$.

C   Flow cytometric analysis of phosphorylated histone H2AX (γH2AX) at the indicated time points after IR (5 Gy, fold versus non-irradiated cells, ctrl). *: one-way ANOVA, $P = 0.03$. $n = 2$.

D   Neutral comet assay performed 24 h after IR (5 Gy). Ctrl: non-irradiated cells. Comet tail's length is proportional to DSB extent.

E   Western blot showing phosphorylation of ATM (pATM) and Chk2 (pChk2) and accumulation of RAD51 after IR (5 Gy). Total ATM protein is also shown. Vinculin was used as a loading control.

F   qPCR of RAD51 expression measured in NS, BT308 diff, and astrocytes at the indicated time points after IR (5 Gy, fold versus non-irradiated cells, ctrl). $n = 3$.

Data information: Data are represented as mean ± SEM.

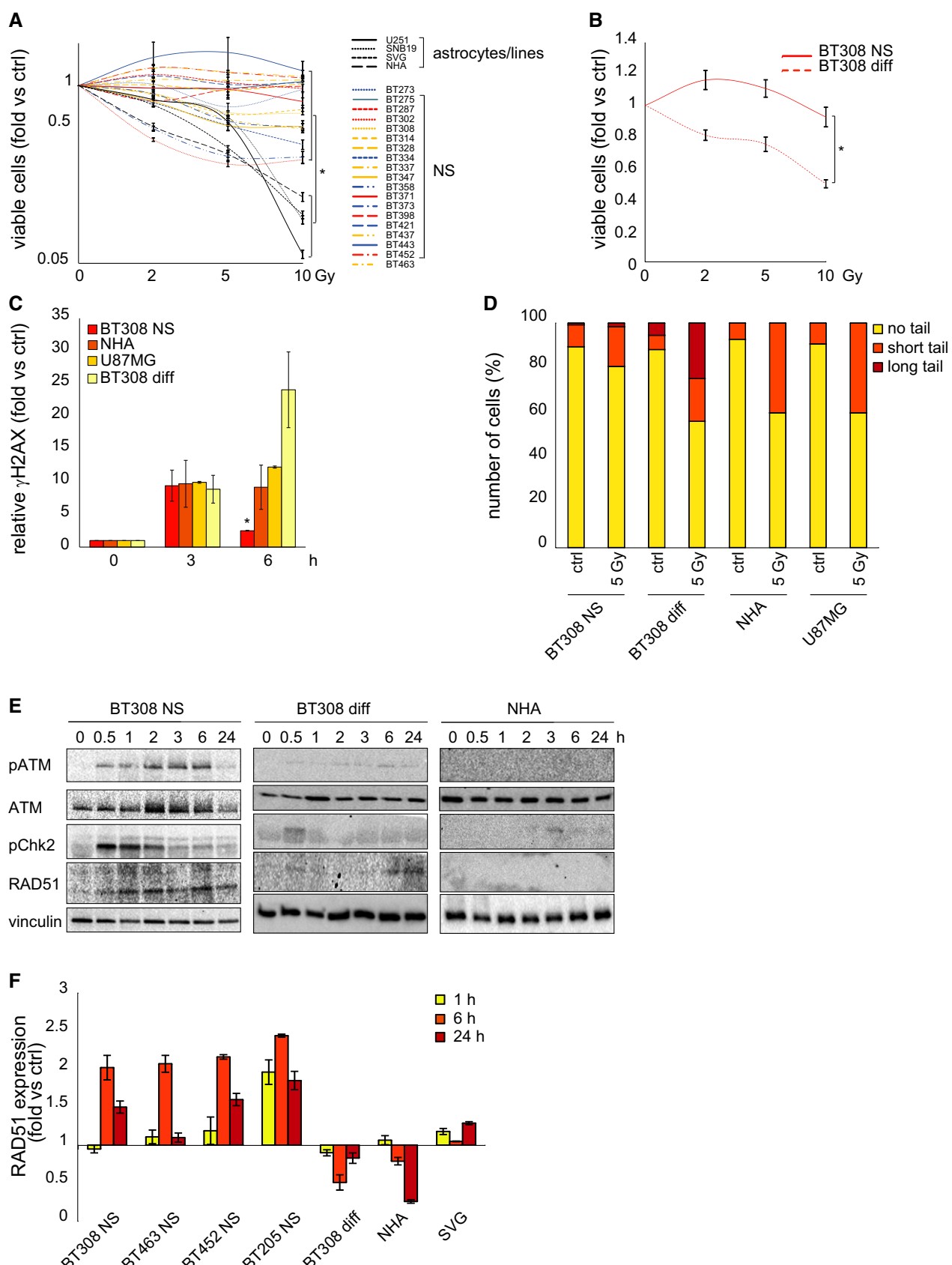

**Figure 1.**

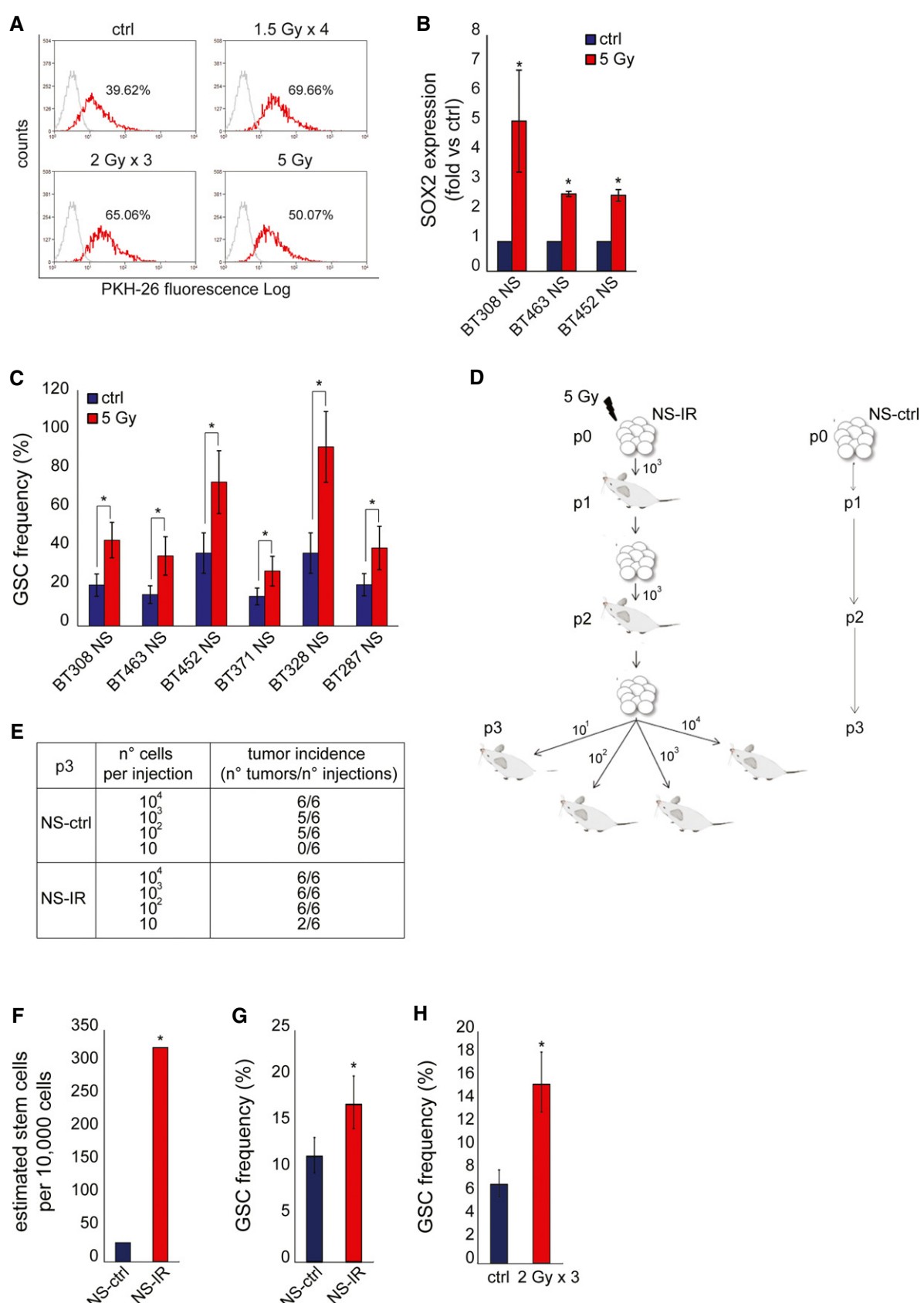

**Figure 2.**

**Figure 2. GSCs are positively selected by irradiation.**

A   Flow cytometric analysis of PKH-26 staining in irradiated BT308NS, 24 h after the last irradiation. The percentage of PKH-26$^{pos}$ cells is indicated. Ctrl: non-irradiated cells. The mean fluorescence intensity (MFI) was as follow: ctrl, 58.21; 1.5 Gy × 4, 69.35; 2 Gy × 3, 68.20; 5 Gy, 60.29.

B   qPCR of SOX2 expression measured in NS 6 h after IR (5 Gy, fold versus non-irradiated cells, ctrl). *: *t*-test, $P < 0.02$. $n = 3$.

C   LDA (sphere-forming) measuring the enrichment of GSCs after IR (5 Gy). Ctrl: non-irradiated cells *: $\chi^2$ test, $P < 0.05$. $n = 2$.

D   Schematic representation of serial xenotransplantation. Control (NS-ctrl) and irradiated (NS-IR) BT308NS were tested for tumor-initiating ability by serial subcutaneous injection of the indicated number of cells (p1, $n = 4$; p2, $n = 6$; p3, $n = 6$ for each condition).

E   Tumor incidence in the LDA performed in p3 mice, as described in (D).

F   Histogram showing the *in vivo* GSC frequency measured by LDA in p3 tumors. GSC frequency (95% CI) was as follows: NS-ctrl, 1/258.2 (97.67-683.9); NS-IR 1/22.1 (8.05-62.6). *: $\chi^2$ test, $P = 1.6 \times 10^{-4}$. $n = 6$.

G   LDA (sphere-forming) measuring the *in vitro* frequency of GSCs in cells derived from p3 tumors. *: $\chi^2$ test, $P = 0.0006$. $n = 3$.

H   LDA (sphere-forming) measuring the *in vitro* frequency of GSCs in cells derived from intracranial tumors generated by BT463NS and irradiated *in vivo* (2 Gy × 3 days) ($n = 3$). Ctrl: non-irradiated tumors. *: $\chi^2$ test, $P = 2.04 \times 10^{-11}$.

Data information: Data are represented as mean ± CI in (C, G, H) or ± SEM in (B).

A panel of NS was stained with PKH-26, showing dye retention in a variable percentage of cells in each NS (on average ~30%) (Fig 2A and Appendix Fig S2A). Next, NS were irradiated with a single (5 Gy) or an equivalent fractionated dose (1.5 Gy × 4 days or 2 Gy × 3 days). After irradiation, the percentage of PKH-26-retaining cells significantly increased, as well as their mean fluorescence intensity (MFI), as compared with non-irradiated cells, indicating that the stem-like NS subpopulation was enriched (Fig 2A and Appendix Fig S2A). A stem-like cell enrichment after NS irradiation was also indicated by (i) qPCR evaluation of SOX2 transcription, previously associated with the GSC phenotype (Fig 2B) (Li *et al*, 2011); (ii) flow cytometric analysis of Olig2, a marker previously used to characterize mesenchymal/proneural radioresistant NS (Appendix Fig S2B) (Bhat *et al*, 2013).

We then showed that irradiation raised the frequency of cells displaying GSC functional properties, as measured by limiting dilution assays (LDA) performed *in vitro* and *in vivo*. In a representative panel of NS, irradiation with 5 Gy significantly increased the frequency of sphere-forming cells (Fig 2C). In a representative NS (BT308NS), irradiation increased the tumorigenic potential. As depicted in Fig 2D, NS were irradiated *in vitro* (NS-IR, p0) and, after 24 h, transplanted subcutis in the mouse (p1). In parallel, an equal number of non-irradiated NS cells (NS-ctrl) were transplanted as control. Both NS-IR and NS-ctrl generated tumors (p1) that were serially passaged by further transplantation of an equal number of

cells (p2). Finally, tumors generated in p2 were passaged as a limiting dilution assay, by transplanting 10–$10^4$ cells in p3 mice. The calculated GSC frequency was ~11-fold higher in tumors originated from NS-IR, as compared with tumors from NS-ctrl (Fig 2E and F). In addition, cells were derived from p3 tumors and assessed in an *in vitro* LDA, showing that the sphere-forming ability significantly increased in cells from tumors that originated from NS-IR, as compared with controls (Fig 2G). In accordance with *in vivo* and *in vitro* evidence of GSC enrichment associated with irradiation, the median volume of tumors generated by NS-IR, comparable to those generated by NS-ctrl at p1, increased through serial passages to a greater extent, as compared with control tumors (Fig EV2A and B). Finally, an increased GSC frequency was also observed in a second GBM model. This tumor was established by intracranic injection of NS, treated *in vivo* with IR (2 Gy × 3 days), and assessed by *in vitro* LDA 62 days after treatment (Fig 2H).

Collectively, these results show that the cell subpopulation endowed with the clonogenic and tumorigenic properties that qualify GSCs is positively selected by IR.

## MET-expressing GSCs are selected by irradiation in experimental models

We have previously shown that (i) MET is expressed in a subset of NS (~40%) sequentially derived from primary GBM (MET-pos-NS);

**Figure 3. MET-expressing GSCs are selected by irradiation.**

A   In MET-pos-NS, the MET$^{high}$ subpopulation retains GSC properties and generates a heterogeneous progeny including also MET$^{neg}$ pseudodifferentiated cells.

B   LDA (sphere-forming) measuring the GSC frequency after IR (5 Gy) in MET$^{high}$ and MET$^{neg}$ subpopulations sorted from BT308NS. *: $\chi^2$ test, $P = 0.001$. $n = 2$.

C   Flow cytometric analysis of MET in irradiated BT308NS, 24 h after the last irradiation (2 Gy × 3 days or a single dose of 5 Gy). Dotted lines: threshold to define the percentage of MET-expressing cells. Ctrl: non-irradiated cells. The MFI was as follows: ctrl, 18; 2 Gy × 3, 22.77; 5 Gy, 19.49.

D   Flow cytometric analysis of MET in BT308NS weekly treated with IR (2 Gy) for 6 weeks. Dotted lines: threshold to define the percentage of MET-expressing cells. Ctrl: non-irradiated cells.

E   Representative immunofluorescence staining of MET (red) on tumors generated by subcutaneous injection of BT308NS irradiated (2 Gy × 3) and explanted 72 h after the last irradiation. Nuclei are counterstained with DAPI (blue). Ctrl: non-irradiated tumors. Scale bar, 10 μm (63× magnification).

F   Left: Quantification of the percentage of MET-expressing cells in tumor sections represented in (E) ($n = 12$ HPF/group). HPF: high-power field. *: *t*-test, $P = 0.001$. Right: Quantification of MET mean fluorescence intensity in tumor sections represented in (E) ($n = 12$ HPF/group, fold versus non-irradiated cells, ctrl). *: *t*-test, $P = 0.00016$.

G   Representative MET immunohistochemical staining of matched primary and recurrent tumors. Scale bar, 50 μm (40× magnification).

H   Histogram representing the number of patient displaying different percentages of MET-expressing cells in their primary or recurrent tumors: negative, 0–5%; low, 6–29%; moderate, 30–69%; high, 70–100% (see also Appendix Table S4).

I   Dot plot representing the MET cumulative score (MET-positive cell score + MET staining intensity score) in 19 primary and recurrent tumors (see Appendix Table S4). *, Wilcoxon, $P < 0.02$.

Data information: Data are represented as mean ± SEM in (F, I) or ± CI in (B).

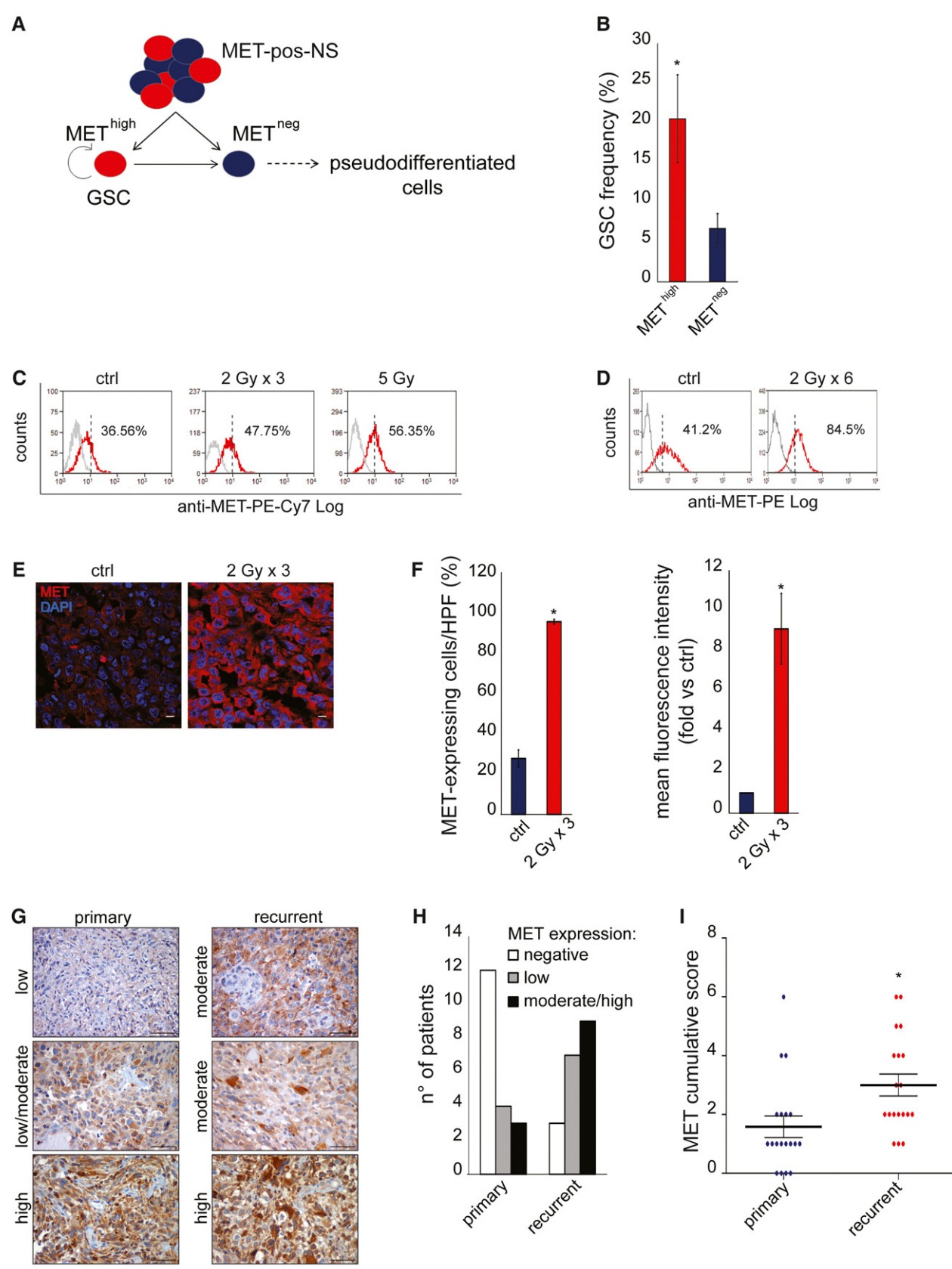

Figure 3.

(ii) MET is a marker of the GSC subpopulation (MET[high]) (De Bacco *et al*, 2012) (Fig 3A); and (iii) MET promotes radioresistance of conventional cancer cell lines (De Bacco *et al*, 2011). Therefore, in the subset of MET-pos-NS (Appendix Table S3), we investigated the relationship between MET expression and GSC radioresistance. Accordingly with our previous data, *in vitro* LDA (sphere-forming assay) showed that the MET[high] subpopulation, sorted from representative MET-pos-NS, was enriched in GSCs (Fig 3B and Appendix Fig S3A). As assessed by flow cytometry, in MET-pos-NS, the number of MET-expressing cells, and their MFI, significantly increased 24 h after irradiation (Fig 3C and Appendix Fig S3B). An even higher enrichment of MET-expressing cells was observed after a chronic IR treatment (Fig 3D). Accordingly, in tumors established by subcutaneous transplantation of MET-pos-NS, the number of MET-expressing cells and the intensity of staining were significantly increased 72 h after the last *in vivo* irradiation (Fig 3E and F).

As MET expression is inducible by IR through NF-κB activation (De Bacco *et al*, 2011), we asked whether MET upregulation could contribute to the enrichment of MET-expressing cells observed after irradiation. Indeed, MET expression was induced by IR in MET[high] cells sorted from MET-pos-NS, but not in the remaining MET[neg] subpopulation (Fig EV3A–D). Lack of MET mRNA accumulation in irradiated MET[neg] cells could be explained by the abundant presence of miRNA targeting MET (such as miR34a and 23b), which was previously associated with this subpopulation (De Bacco *et al*, 2012).

Collectively, these data indicate that MET-expressing GSCs are radioresistant, likely as a result of concomitant mechanisms, involving both cell-positive selection and MET transcriptional upregulation.

## MET-expressing cells are enriched in recurrent human GBMs

We asked whether radioresistance of MET-expressing GSCs could operate in patients' tumors and be associated with positive selection under therapeutic pressure. We collected a panel of 19 pairs of GBMs, each including a primary tumor removed by surgery without prior treatment, and the matched recurrent tumor, arising after adjuvant chemoradiotherapy, and again surgically resected (Appendix Table S4). Remarkably, immunohistochemical analysis of whole tissues showed that significant MET expression (MET cumulative score ≥ 2) was detectable in 36% of primary tumors (7/19), but in 84% of recurrent tumors (16/19) (Appendix Table S5). In the majority of cases, the percentage of MET-positive cells and/or the intensity of MET staining (summed in the cumulative score) increased in the recurrent as compared with the matched primary tumor (Fig 3G–I and Appendix Table S5).

These data suggest that, in primary GBMs, cells expressing MET, being resistant to the standard therapeutic regimen invariably including radiotherapy, can drive tumor recurrence. Although we do not provide functional proofs, we reason that the MET-positive cell population expanded in the recurrent tumor can largely correspond to stem-like cells because: (i) In MET-pos-NS, the MET-negative cell subpopulation is devoid of stem properties, as previously shown (De Bacco *et al*, 2012), and further investigated by *in vitro* LDA (Fig 3B and Appendix Fig S3A); and (ii) GSC differentiation is characterized by loss of MET expression, as shown *in vitro* (De Bacco *et al*, 2012).

The limited size of this cohort did not allow to establish statistically significant correlations between MET expression and clinical outcomes such as disease-free or overall survival (Appendix Table S4). However, in the TCGA dataset, encompassing 401 GBMs analyzed by whole tissue transcriptional profiling (Cerami *et al*, 2012; Gao *et al*, 2013), high MET expression in the primary tumor was significantly associated with decreased disease-free and overall survival (Appendix Fig S3C and D).

## MET[high] GSCs efficiently activate DDR

Radioresistance of the MET-expressing stem-like subpopulation was further assessed by several experimental approaches. First, in irradiated MET-pos-NS, the cell death response was evaluated by measuring Annexin V incorporation by flow cytometry (Fig 4A). About 36 h after IR, the MET-expressing subpopulation (red) mostly contained viable cells (Annexin V/DAPI-negative cells, ~70%). Conversely, the MET-negative subpopulation (blue) displayed a significant decrease in viable cells as compared with non-irradiated controls (~25% versus 74%), and a cell redistribution in early and, mostly, late apoptotic phases (Annexin V[pos]/DAPI[neg]: early apoptotic; Annexin V[pos]/DAPI[pos]: late apoptotic cells) (Fig 4A). Cell death of the MET-negative subpopulation was likely due to apoptosis, as indicated by analysis of PARP and caspase-3 activation, which was evident in sorted and irradiated MET[neg] cells, but not in MET[high] cells (Fig 4B).

The biological response to IR was further evaluated in sorted MET[high] and MET[neg] cells. Viability, measured in dose–response experiments 24 h after IR, was significantly higher in MET[high] than in MET[neg] cells at all doses tested (Fig 4C). Consistently, in radiobiological clonogenic assays, the surviving fraction of MET[high] cells remained almost unchanged even after irradiation with the highest dose (10 Gy), while that of MET[neg] cells significantly decreased, tending to exhaust after 10 Gy (Fig 4D and Appendix Fig S4A–C).

Finally, in comet assays performed 24 h after irradiation, MET[high] cells displayed modest DSB accumulation, as compared with non-irradiated cells, while MET[neg] displayed severe damage (Fig 4E and Appendix Fig S4D and E).

The radioresistance of the MET[high] subpopulation was then correlated with constitutive DDR activation, as previously assessed in GSCs prospectively isolated with the CD133 marker (Bao *et al*, 2006). Indeed, untreated MET[high], but not MET[neg] cells, displayed constitutive phosphorylation of ATM and Chk2 kinases. After irradiation, ATM phosphorylation was significantly increased in MET[high] cells, but only weakly induced in MET[neg] (Fig 4F and Appendix Fig S4F). Consistently, RAD51 expression was significantly more abundant in irradiated MET[high] than MET[neg] cells (Fig 4G). Accordingly, 24 h after irradiation, histone H2AX phosphorylation was still detectable only in MET[neg] cells (Fig 4F).

Collectively, these findings indicate that the positive selection of MET[high] GSC by IR resides on intrinsic radioresistance of this subpopulation, likely conferred by enhanced basal activity, and efficient hyperactivation of DDR effectors after irradiation.

## MET inhibition radiosensitizes GSCs

The above results indicate that MET is a marker of a radioresistant GSC subpopulation. We thus investigated whether MET signaling

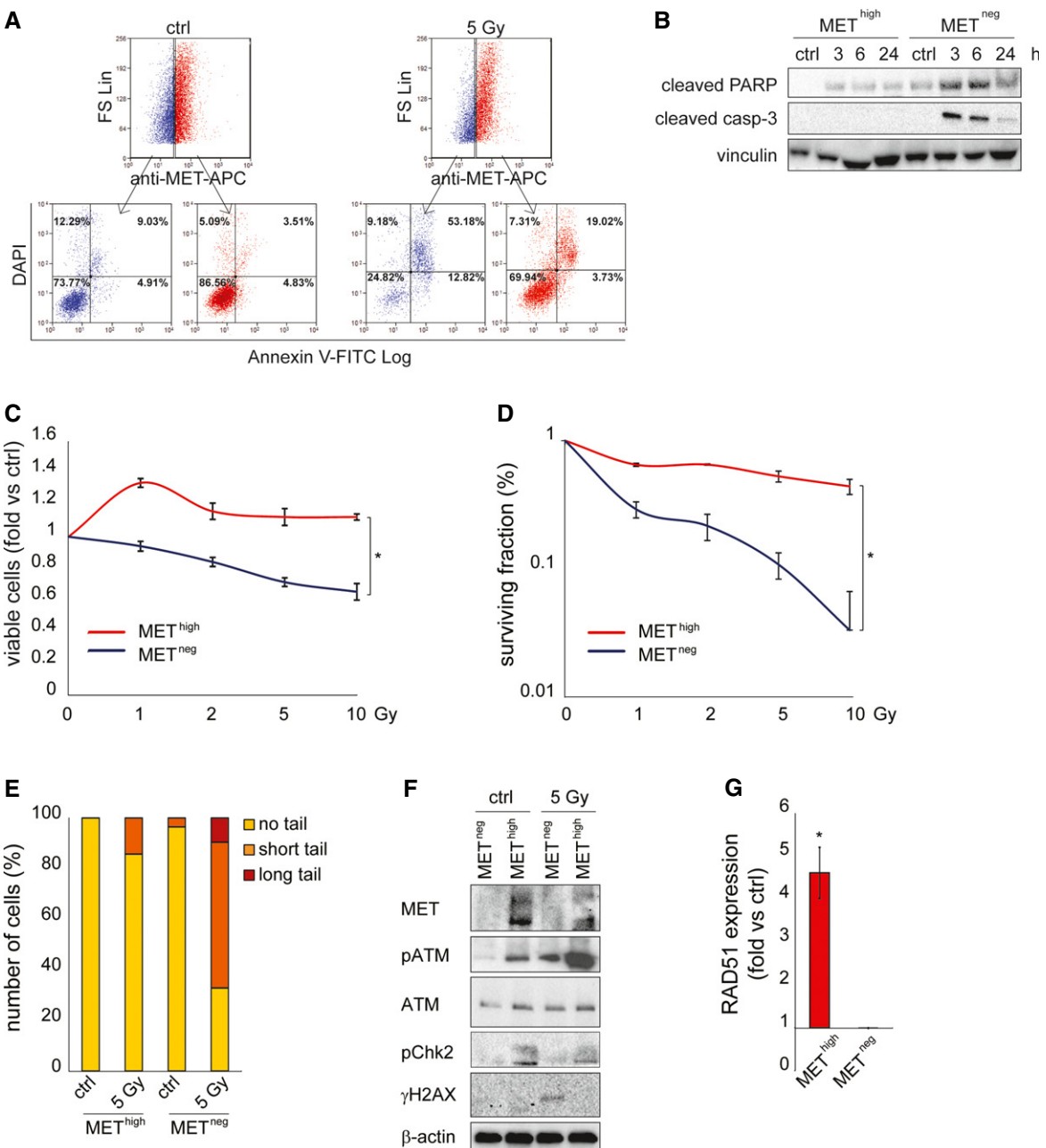

**Figure 4. MET^high GSCs efficiently activate DDR.**

A   Flow cytometric analysis of Annexin V/DAPI incorporation and MET expression in BT308NS 36 h after IR (5 Gy). Ctrl: non-irradiated cells. Blue: MET-negative cells. Red: MET-expressing cells.

B   Western blot of MET^neg and MET^high subpopulations sorted from BT452NS, showing activation of PARP (cleaved PARP) and caspase-3 (cleaved casp-3) after IR (5 Gy). Vinculin was used as a loading control.

C   Cell viability of MET^high and MET^neg subpopulations sorted from BT308NS, measured 96 h after IR (1–10 Gy, fold versus non-irradiated cells, ctrl). *: t-test, P < 0.001. n = 2.

D   Clonogenic assay with MET^high and MET^neg subpopulations sorted from BT308NS and irradiated (1–10 Gy, fold versus non-irradiated cells, ctrl). *: t-test, P < 0.05. n = 3.

E   Neutral comet assay in MET^high and MET^neg subpopulations sorted from BT308NS, performed 24 h after IR (5 Gy). Ctrl: non-irradiated cells. Comet tail's length is proportional to DSB extent.

F   Western blot of MET^neg and MET^high subpopulations sorted from BT308NS, showing constitutive (ctrl) and/or IR-induced phosphorylation of ATM (pATM), Chk2 (pChk2), and H2AX (γH2AX), and accumulation of RAD51, 24 h after IR (5 Gy). Total ATM protein is also shown. β-actin was used as a loading control.

G   qPCR of RAD51 expression in MET^high and MET^neg subpopulations sorted from BT308NS 24 h after IR (5 Gy, fold versus non-irradiated control cells, ctrl). *: t-test, P = 0.02. n = 3.

Data information: Data are represented as mean ± SEM.
Source data are available online for this figure.

**Figure 5.  MET inhibition radiosensitizes GSCs.**

A   Cell viability of BT308NS, measured 48 h after IR (2 or 5 Gy) in the absence (mock) or in the presence of 10 ng/ml HGF (fold versus non-irradiated mock treated cells, mock). Ctrl: non-irradiated cells. *: *t*-test (5 Gy + HGF) versus (5 Gy), *P* < 0.0001. *n* = 2.

B   Flow cytometric analysis of Annexin V/DAPI incorporation in BT308NS irradiated (5 Gy) in the absence (mock) or in the presence of HGF. Histograms show the percentage of Annexin V-positive cells 48 h after IR. Ctrl: non-irradiated cells.

C   Cell viability of BT308NS, irradiated in the absence (5 Gy) or in the presence (5 Gy + JNJ) of the MET inhibitor JNJ38877605 (500 nM) and analyzed at the indicated time points. Vehicle: non-irradiated cells (fold versus vehicle-treated cells at time 0, mock). *: *t*-test, *P* < 0.01. *n* = 2.

D   Cell viability as in C, measured 5 days after IR (1–10 Gy, fold versus vehicle-treated cells, mock). *: *t*-test, *P* < 0.03. *n* = 2.

E   Flow cytometric analysis of Annexin V/DAPI incorporation in BT308NS irradiated (5 Gy) in the absence (vehicle) or in the presence (JNJ) of JNJ38877605. Histograms show the percentage of Annexin V-positive cells 36 h after IR. Ctrl: non-irradiated cells. *: *t*-test (5 Gy + JNJ) versus (5 Gy), *P* = 0.01. *n* = 3.

F   Western blot of BT308NS showing caspase-3 activation (cleaved casp-3), at the indicated time points after IR in the absence (−) or in the presence (+) of JNJ38877605. Vinculin was used as a loading control.

G   Clonogenic assay with MET^neg and MET^high subpopulations sorted from BT308NS, irradiated (5 Gy) in the absence (vehicle) or in the presence of MET inhibitors (JNJ38877605 and Crizotinib, 500 nM). Ctrl: non-irradiated cells. *: *t*-test, *P* < 0.0004. *n* = 3.

H   LDA (sphere-forming assay) with MET^neg and MET^high subpopulations sorted from BT308NS, measuring GSC frequency after irradiation in the absence (5 Gy) or in the presence (5 Gy + JNJ) of JNJ38877605. Vehicle: non-irradiated vehicle-treated cells. *: $\chi^2$ test, *P* = 0.0002. *n* = 2.

I   Flow cytometric analysis of phosphorylated histone H2AX (γH2AX) in the whole BT308NS at the indicated time points after IR in the absence (5 Gy) or in the presence (5 Gy + JNJ) of JNJ38877605 (fold versus non-irradiated cells at time 0, mock). *: *t*-test, *P* < 0.05. *n* = 2.

J   Neutral comet assay with BT308NS performed 24 h after IR in the absence (5 Gy) or in the presence (5 Gy + JNJ) of JNJ38877605. Vehicle: non-irradiated vehicle-treated cells. Comet tail's length is proportional to DSB extent.

Data information: Data are represented as mean ± SEM in (A, C–E, G, I) or ± CI in (H).
Source data are available online for this figure.

has a functional role in sustaining radioresistance, and whether its inhibition can radiosensitize GSCs. MET-pos-NS were kept in a standard EGF/bFGF medium, supplied with HGF to better mimic the brain (tumor) microenvironment (Kunkel *et al*, 2001; Xie *et al*, 2012). In this condition, after irradiation, MET gets hyperphosphorylated (Fig EV4A and B), consistently with previous results obtained in cell lines (Lal *et al*, 2005; De Bacco *et al*, 2011). Addition of HGF to standard culture medium increased radioresistance of irradiated MET-pos-NS, by preserving viability (Fig 5A and Appendix Fig S5A and B) and preventing apoptosis (Fig 5B). We then showed that association of the specific MET inhibitor JNJ38877605 to IR significantly reduced cell viability and increased the percentage of apoptotic (Annexin V-positive) cells, as compared with treatment with IR alone (Fig 5C–E and Appendix Fig S5C and D). Consistently, caspase-3 activation was barely detectable in cells treated with IR alone, to become well evident in those treated with IR and the MET inhibitor (Fig 5F). The effect of MET targeting was also assessed by the use of alternative inhibitors, including the MET-specific small-molecule PHA665752, the monovalent form of the MET monoclonal antibody DN30 (MvDN30) (Pacchiana *et al*, 2010), and the promiscuous, clinically approved, small-molecule Crizotinib (targeting MET and ALK kinases). All the above inhibitors combined with IR similarly reduced cell viability and increased caspase-3/7 activation, as compared with IR alone (Appendix Fig S5E–H). These outcomes, observed in whole NS, were obviously due to the activity of the inhibitors in MET-expressing GSCs and not in the MET-negative non-stem subpopulation, which, by definition, lacks the target. Consistently, in radiobiological clonogenic assays performed with the sorted MET^neg and MET^high subpopulations, the surviving fraction of MET^neg cells treated with IR, already very low, was not further decreased by MET inhibitors. Conversely, the surviving fraction of the irradiated MET^high subpopulation was reduced by MET inhibitors to the levels of the MET^neg cells (Fig 5G and Appendix Fig S5I). Finally, in LDA (sphere-forming assay) with sorted MET^high and MET^neg subpopulations, the GSC frequency of the MET^high subpopulation was abated by combination of IR with the MET inhibitor, while, as expected, the low GSC frequency of the

MET^neg subpopulation was further decreased only by IR (Fig 5H). Interestingly, a significant decrease of MET^high GSC frequency was also observed after treatment with the MET inhibitor alone, indicating that HGF present in the culture medium plays a role in sustaining self-renewal, as previously shown (Li *et al*, 2011; De Bacco *et al*, 2012) (Fig 5H). However, no apoptosis was expected, and indeed observed, by treating NS with MET inhibitors alone (Fig 5E and F and Appendix Fig S5E–H).

The radiosensitizing activity of MET inhibitors was then correlated with the ability to prevent DNA repair. In whole MET-pos-NS, treatment with IR and the MET inhibitor JNJ38877605 increased the levels of histone H2AX phosphorylation and the extent of DSBs, as compared with IR alone (Fig 5I and J and Appendix Fig S5J and K).

These findings show that MET behaves as a functional marker of radioresistance, whose inhibition efficiently radiosensitizes MET^high GSCs.

**MET inhibition impairs GSC DDR by preventing AKT-dependent ATM and p21 activity**

We thus asked whether MET inhibition directly impinges on the molecular mechanisms regulating DDR. We previously showed that, after irradiation of conventional cell lines, ATM upregulates MET expression through NF-κB transcription factor (De Bacco *et al*, 2011). We now show that MET, in turn, unleashes a signal transduction cascade that sustains ATM activity and DDR. We found that, in irradiated MET-pos-NS kept in the standard EGF/bFGF medium, addition of HGF increased ATM and Chk2 phosphorylation, and RAD51 expression (Fig 6A and Appendix Fig S6A). Conversely, in irradiated MET-pos-NS kept in the medium supplied with HGF, addition of the MET inhibitor significantly reduced ATM and Chk2 phosphorylation, and RAD51 expression (Fig 6B and C). This effect depended specifically on MET inhibition, as shown by complementary experimental approaches such as: (i) the use of additional MET inhibitors such as the above-described PHA665752, Crizotinib, and MvDN30 (Appendix Fig S6B–D), which invariably

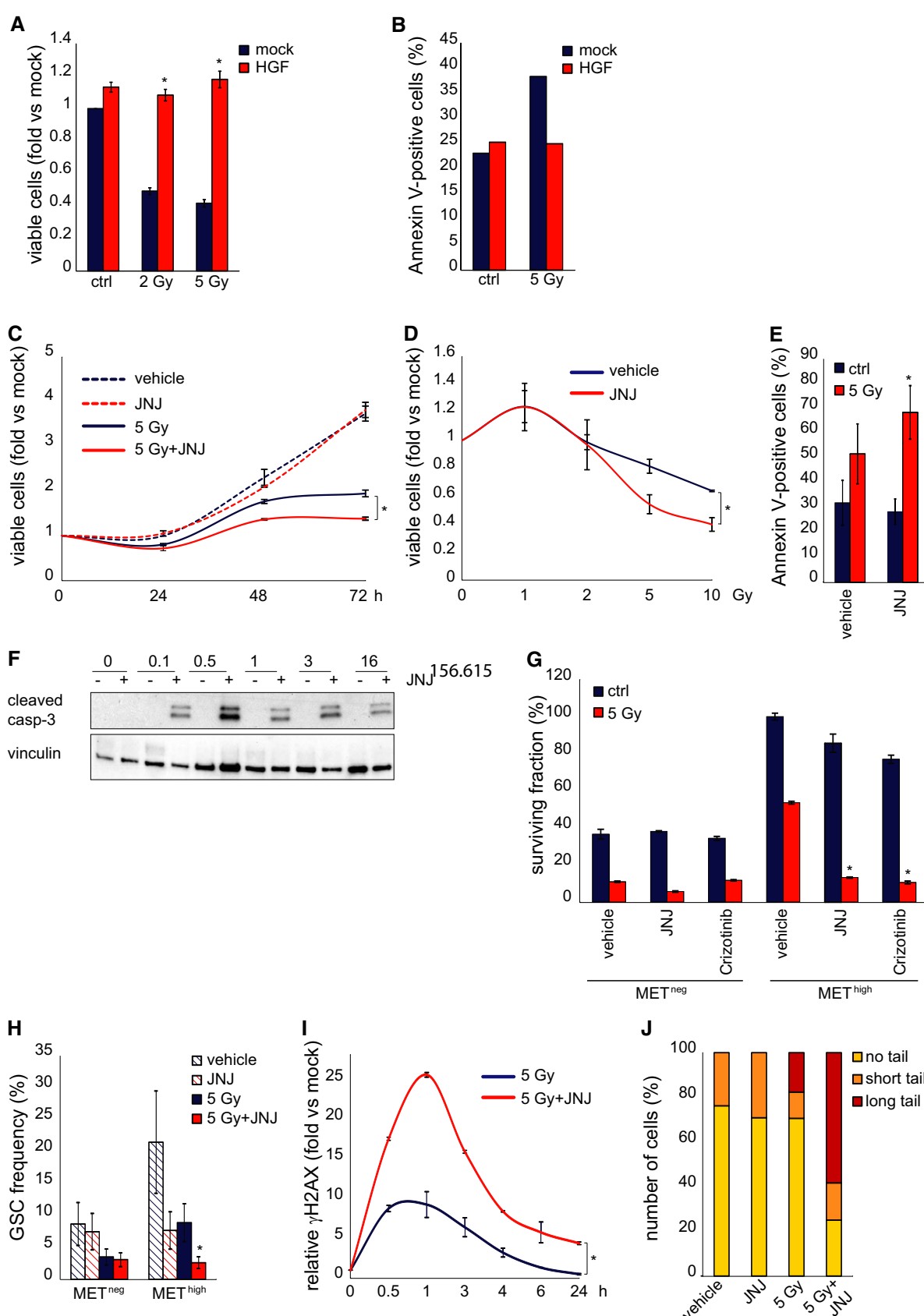

**Figure 5.**

**Figure 6.  MET inhibition impairs AKT-dependent ATM activity.**

A   Western blot of BT463NS showing phosphorylation of ATM (pATM) and Chk2 (pChk2), and accumulation of RAD51 at the indicated time points after IR in the absence (5 Gy) or in the presence (5 Gy + HGF) of HGF (10 ng/ml). Total ATM is also shown. H3 was used as a loading control.

B   Western blot of BT308NS showing phosphorylation of ATM (pATM) and Chk2 (pChk2) and accumulation of RAD51 at the indicated time points after IR in the absence (5 Gy) or in the presence (5 Gy + JNJ) of the MET inhibitor JNJ38877605 (500 nM). Total ATM is also shown. β-actin was used as a loading control. Veh: non-irradiated vehicle-treated cells.

C   Densitometric analysis of ATM phosphorylation shown in (B). *: *t*-test, *P* < 0.01. *n* = 3.

D   Western blot of BT308NS showing phosphorylation of ATM, Chk2, and AKT (pAKT) and accumulation of RAD51 24 h after IR (5 Gy) in the absence (vehicle, veh) or in the presence of MAPK (PD98059, 20 μM), AKT (Ly294002, 20 μM), MET (JNJ38877605), or ATM (CGK733, 10 μM) inhibitors. Total ATM and AKT are also shown. β-actin was used as a loading control. Ctrl: non-irradiated cells.

E   Western blot of BT463NS showing phosphorylation of ATM, Chk2, and Aurora kinase A (pAurora A) at the indicated time points after IR in the absence (5 Gy) or in the presence (5 Gy + Ly) of the AKT inhibitor Ly294002. Total ATM and Aurora kinase A are also shown. H3 was used as a loading control. Veh: non-irradiated vehicle-treated cells.

F   Western blot of BT308NS showing phosphorylation of Aurora kinase A at the indicated time points after IR in the absence (5 Gy) or in the presence (5 Gy + JNJ) of JNJ38877605. Total Aurora kinase A is also shown. H3 was used as a loading control. Veh: non-irradiated vehicle-treated cells.

G   Western blot of BT463NS showing phosphorylation of Aurora kinase A at the indicated time points after IR in the absence (5 Gy) or in the presence (5 Gy + HGF) of HGF. Total Aurora kinase A is also shown. H3 was used as a loading control (the same control applies to both panel A and G).

H   Western blot of BT308NS showing phosphorylation of ATM and Aurora kinase A at the indicated time points after IR in the absence (5 Gy) or in the presence (5 Gy + MLN) of the Aurora kinase A inhibitor MLN8237 (20 μM). Total ATM and Aurora kinase A are also shown. H3 was used as a loading control. Veh: non-irradiated vehicle-treated cells.

I   Cell viability of BT308NS, measured 72 h after IR (5 Gy) in the absence (vehicle) or in the presence of MET (JNJ38877605), Aurora kinase A (MLN8237), or AKT (Ly294002) inhibitors (fold versus non-irradiated vehicle-treated cells, mock). *: *t*-test (5 Gy + inhibitor) versus (5 Gy + vehicle), *P* < 0.01. *n* = 2.

Data information: Data are represented as mean ± SEM.
Source data are available online for this figure.

decreased ATM phosphorylation in irradiated MET-pos-NS; and (ii) the use of a MET inhibitor in MET-neg-NS (lacking MET protein expression, Appendix Table S3), which failed to impair ATM phosphorylation, thus ruling out off-target effects (Appendix Fig S6E).

We then investigated the mechanistic link between MET and ATM phosphorylation. In MET-pos-NS, IR induced activation of MET (see above, Fig EV4A and B) and its two main downstream signaling pathways, AKT and MAP kinases (MAPK). Phosphorylation of both AKT and MAPK was dependent of MET, as it was fully prevented by combining IR with the MET inhibitor (Appendix Fig S6F). We further showed that AKT was responsible to connect MET to ATM, as an AKT—but not a MAPK—inhibitor decreased ATM phosphorylation, and activation of its downstream effectors Chk2 and RAD51 after irradiation (Fig 6D and Appendix Fig S6G). The pathway linking AKT to ATM phosphorylation is currently unknown. A likely candidate to investigate was Aurora kinase A, known to phosphorylate ATM (Yang *et al*, 2011). Indeed, in irradiated MET-pos-NS, we showed that (i) Aurora kinase A phosphorylation on Thr288, required for enzymatic activation (Walter *et al*, 2000), was prevented by the AKT inhibitor LY294002 (Fig 6E and Appendix Fig S6H), as well as by the MET inhibitor (Fig 6F and Appendix Fig S6I); (ii) Aurora kinase A phosphorylation was increased by HGF (Fig 6G and Appendix Fig S6J); and (iii) Aurora kinase A inhibition by its specific small-molecule inhibitor MLN8237 prevented ATM phosphorylation (Fig 6H and Appendix Fig S6K). Consistently, AKT or Aurora kinase A inhibitors mimicked the radiosensitizing activity of the MET inhibitor, as shown in viability experiments (Fig 6I and Appendix Fig S6L and M).

The role of the AKT pathway as a crucial player of MET-induced radioresistance was further supported by p21 analysis. It is known that, when it is phosphorylated by AKT on Thr145, p21 is retained in the cytoplasm and exerts anti-apoptotic functions; in the absence of this phosphorylation, p21 translocates into the nucleus to block the cell cycle (Abbas & Dutta, 2009). Indeed, we showed that, in irradiated MET-pos-NS kept in the standard EGF/bFGF medium,

addition of HGF induced p21 phosphorylation on Thr145 (Fig 7A) and its cytoplasmic accumulation (Fig 7A). In a complementary experiment, in irradiated MET-pos-NS kept in the medium supplied with HGF, addition of the MET inhibitor prevented p21 phosphorylation on Thr145 (Fig 7B and Appendix Fig S7A), and induced its nuclear translocation, as shown by Western blot and immunofluorescence analysis (Fig 7C–E and Appendix Fig S7B and C). Consistently, when MET-pos-NS were irradiated in the presence of the MET inhibitor, cell cycle was arrested in the G2-M phase (Fig 7F and Appendix Fig S7D) and apoptosis was unleashed (see above, Fig 5E and F and Appendix Fig S5E–H).

Collectively, these data indicate that MET supports DDR via AKT activation, which, on its turn, sustains the ATM pathway for DNA repair, and p21 cytoplasmic retention, connected with anti-apoptotic functions (Fig 7G). Impairment of these signaling circuits provides a mechanistic explanation for the radiosensitizing effect of MET inhibitors.

**MET inhibition radiosensitizes GSC-derived GBMs by GSC depletion**

We then generated GBMs by *in vivo* transplantation of MET-pos-NS, to investigate whether combination with MET inhibitors could increase the efficacy of radiotherapy by contributing to deplete GSCs. As assessed, the MET inhibitor JNJ38877605 crosses the blood–brain barrier (Appendix Fig S8A). GBMs were then established by intracranial xenotransplantation of BT463NS. Ten days after NS injection, mice were randomized into four treatment groups: (i) vehicle, (ii) IR (2 Gy × 3 days), (iii) JNJ38877605, supplied for 30 days, and (iv) combination therapy (combo, IR and JNJ38877605 as above). Approximately 60 days after the beginning of treatment, at the onset of severe neurological symptoms in controls, mice were sacrificed and brains were analyzed by epifluorescence imaging (Fig 8A). Combination therapy dramatically reduced tumor growth, measured as GFP intensity, as compared with IR or MET inhibitor alone (Fig 8B). In a second model, tumors were established by subcutaneous

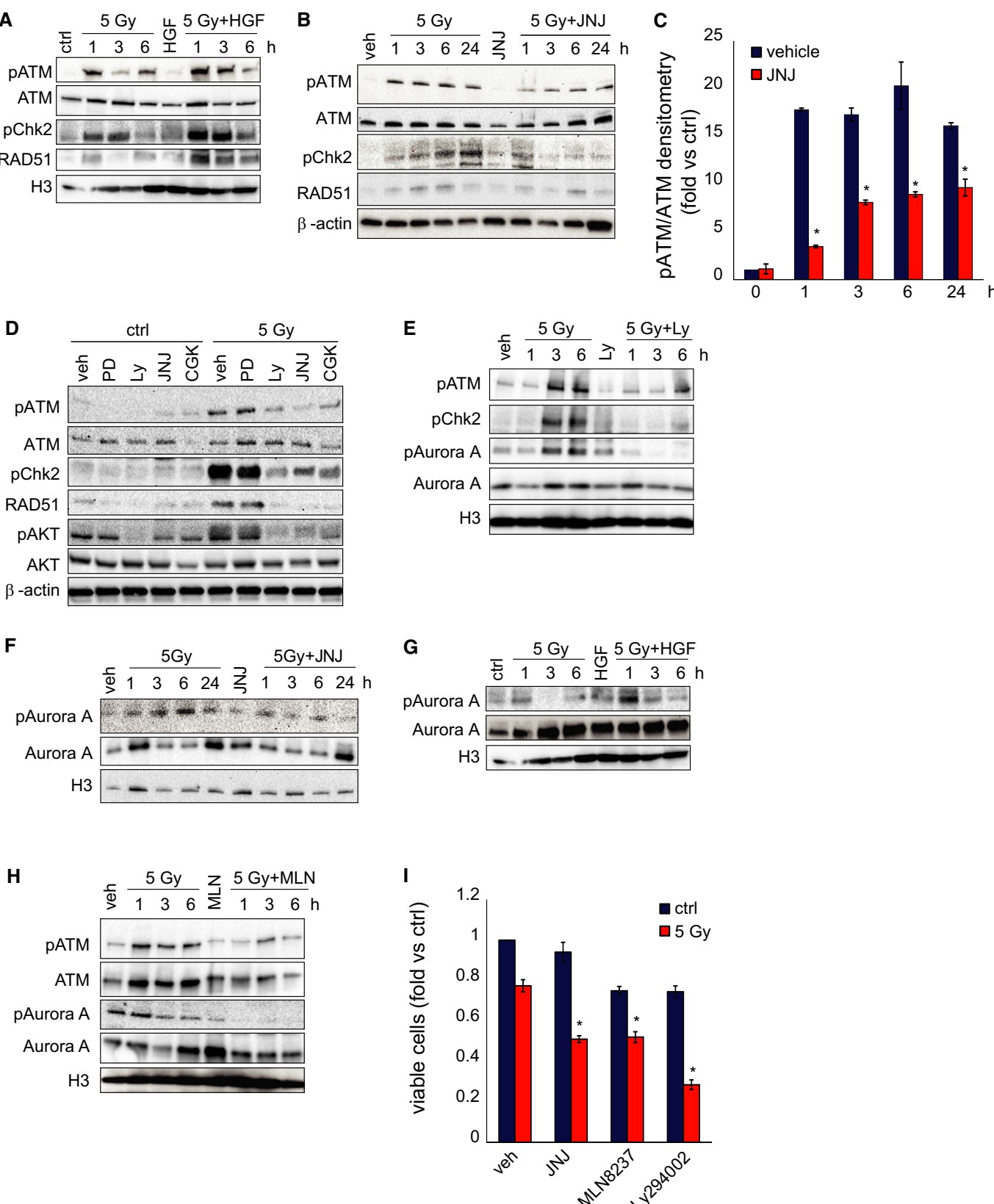

**Figure 6.**

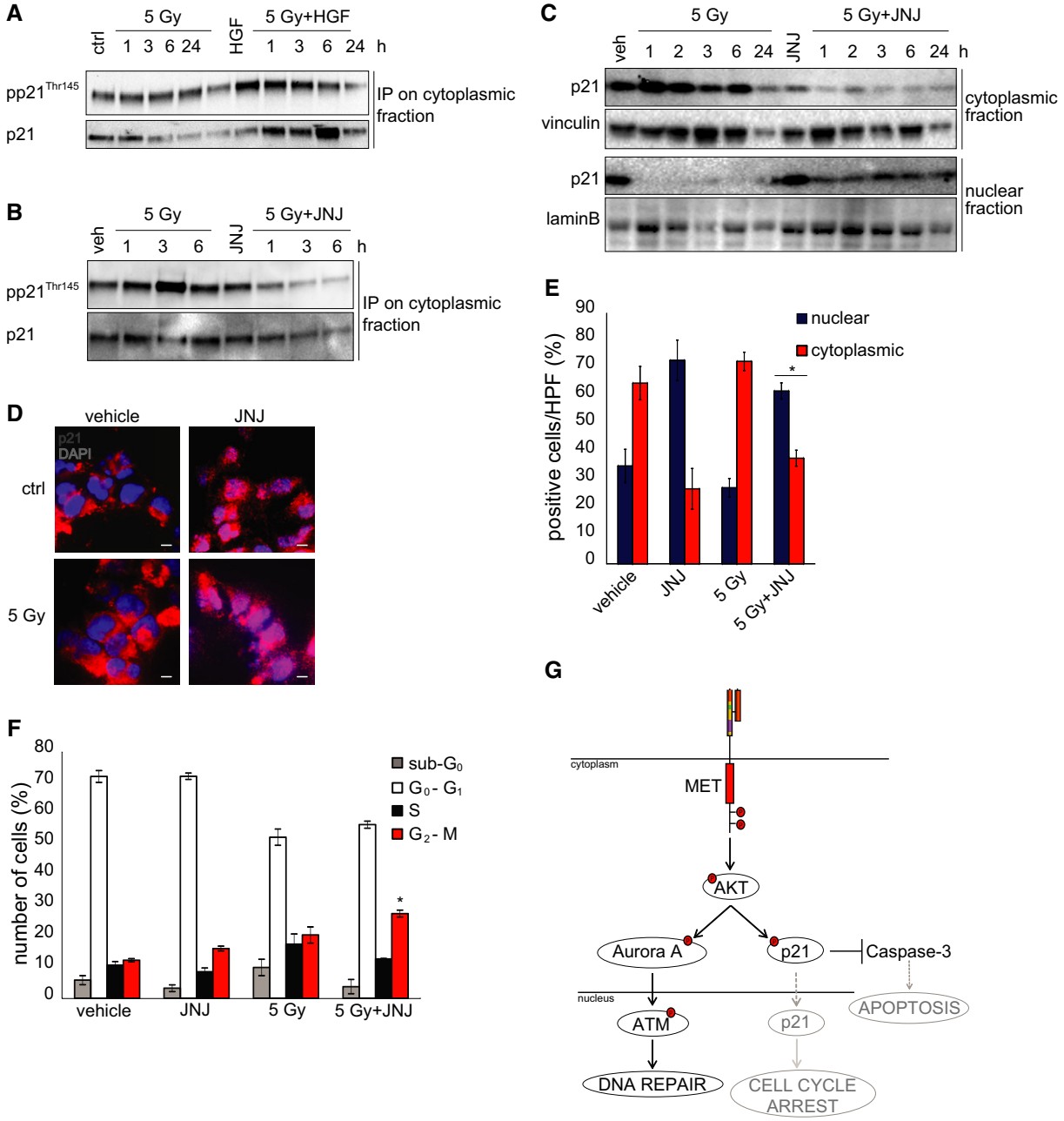

**Figure 7. MET inhibition promotes p21 nuclear translocation.**

A   Western blot of BT371NS showing p21 phosphorylation (pp21$^{Thr145}$) in cytoplasmic lysates immunoprecipitated with p21 antibodies at the indicated time points after IR in the absence (5 Gy) or in the presence (5 Gy + HGF) of HGF (10 ng/ml).

B   Western blot of BT463NS showing p21 phosphorylation (pp21$^{Thr145}$) in cytoplasmic lysates immunoprecipitated with p21 antibodies at the indicated time points after IR in the absence (5 Gy) or in the presence (5 Gy + JNJ) of the MET inhibitor JNJ38877605 (500 nM).

C   Western blot of BT308NS showing p21 localization in cytoplasmic/nuclear extracts at the indicated time points after IR in the absence (5 Gy) or in the presence (5 Gy + JNJ) of JNJ38877605. Vinculin and lamin B were used as loading controls for cytoplasmic and nuclear fractions, respectively.

D   Representative immunofluorescence staining of p21 (red) on BT308NS cells 3 h after IR (5 Gy) in the absence (vehicle) or in the presence (JNJ) of JNJ38877605. Ctrl: non-irradiated cells. Nuclei are counterstained with DAPI (blue). Scale bar, 10 μm (63× magnification).

E   Quantification of the percentage of cells showing p21 cytoplasmic or nuclear localization in BT308NS represented in (D) (n = 10 HPF/group). HPF: high-power field. *: t-test (5 Gy + JNJ) versus (ctrl or 5 Gy), P < 0.0001.

F   Cell cycle analysis of BT314NS, performed 24 h after IR in the absence (5 Gy) or in the presence (5 Gy + JNJ) of JNJ38877605. Vehicle: non-irradiated cells. *: one-way ANOVA, P = 0.008. n = 3.

G   Schematic representation of the MET-driven signaling pathways that sustain DNA repair and prevent apoptosis and cell cycle arrest.

Data information: Data are represented as mean ± SEM.

Source data are available online for this figure.

transplantation of BT308NS, displaying a GBM-like morphology (Appendix Fig S8B), and were treated as above (JNJ38877605 was administered for 15 days). Tumors treated with vehicle, or the MET inhibitor alone, reached the experimental endpoint (tumor volume = 1,600 mm$^3$) within 18 or 32 days, respectively. Tumors treated with IR alone remained stable for 40 days after beginning of the treatment, leading to the experimental endpoint within 63 days. Remarkably, tumors treated with the combination therapy showed clinical regression (volume reduction > 50% as compared with day 0), which persisted until 40 days, leading to the experimental endpoint 90 days after beginning of the treatment (Fig 8C and D). A significant growth inhibition (~2-fold) by combination therapy as compared with radiotherapy alone was observed also in tumors generated by transplantation of BT371NS, although these tumors were not arrested by any therapy (Appendix Fig S8B–D).

Next, we investigated whether combination treatment led to decreased GSC frequency. As depicted in Fig 8E, in an independent experiment, tumors were generated by BT308NS subcutaneous transplantation, treated as above, explanted 10 days after beginning of the treatment, that is, when regression was observed (Appendix Fig S8E), and analyzed by LDA. Both *in vitro* clonogenic (Fig 8F) and *in vivo* tumorigenic (Fig 8G–I) LDA indicated a significant GSC frequency decrease in tumors treated with combination therapy, as compared with controls or radiotherapy (or the MET inhibitor) alone. Of note, *in vitro* but not *in vivo* LDA (p1) allowed to detect an increased GSC frequency after radiotherapy alone. This is expected, as *in vivo* LDA, measuring tumorigenic potential, requires at least two serial passages (p2) to highlight positive selection of GBM stem-like cells by irradiation, as shown above (Figs 2F and EV2A and B). However, in this experiment, further passaging was prevented by the minimal volume, and poor cell viability, of tumors generated by the first passage (p1) after combination therapy (Appendix Fig S8F).

Collectively, these data indicate that radiosensitization of GBM by MET inhibitors not only impairs tumor growth, but also depletes the GSC subpopulation responsible for tumor generation and recurrence.

## Discussion

Radioresistance is a well-known pitfall of GBM therapy (Squatrito & Holland, 2011). The responsible cell subpopulation(s), as well as the underlying genetic and molecular mechanisms, are still poorly understood. Pioneering work has associated radioresistance of GBM and other tumors with the cancer stem cell phenotype, particularly with the intrinsic ability to efficiently activate DDR (Bao *et al*, 2006; Phillips *et al*, 2006b; Diehn *et al*, 2009; Pajonk *et al*, 2010; Cheng *et al*, 2011; Vlashi & Pajonk, 2014; Ahmed *et al*, 2015). In GBM, this property is likely inherited from the cell of origin, a neural stem/progenitor cell (Blanpain *et al*, 2011). More recently, GSC radioresistance has been associated with the GBM mesenchymal profile, implying a causal role for the subtype-specific genetic lesions and/or signaling circuits (Bhat *et al*, 2013). The latter findings are consistent with the evidence that the GBM mesenchymal profile and the epithelial–mesenchymal transition in general overlap with the stem cell phenotype (Phillips *et al*, 2006a; Scheel & Weinberg, 2012) and thus with inherent radioresistance. In the panel of twenty NS analyzed in this study, radioresistance was found to be a common property, displayed by mesenchymal as well as by classical or proneural NS. In classical NS, the typical EGFR amplification could play a prominent role in conferring radioresistance, as indicated by previous studies (Squatrito & Holland, 2011). In mesenchymal and proneural NS, usually lacking EGFR amplification, we investigated whether a prominent role could be played by the wild-type MET oncogene, which was previously associated with the GBM mesenchymal profile (Phillips *et al*, 2006a; Verhaak *et al*, 2010; De Bacco *et al*, 2012), and is usually expressed in a mutually exclusive fashion with EGFR (Verhaak *et al*, 2010; De Bacco *et al*, 2012).

Building on previous studies showing that MET is a marker of a GSC subtype, which identifies and sustains the stem phenotype (Li *et al*, 2011; De Bacco *et al*, 2012; Joo *et al*, 2012), we now show that MET is a functional marker of GSC radioresistance. Indeed, we found that IR positively selects MET$^{high}$ GSCs and that the radioresistant phenotype of these cells correlates with high basal expression levels of ATM, Chk2, and RAD51, which likely prime cells to efficiently activate DDR. As a result, MET$^{high}$ GSCs, but not MET$^{neg}$ non-stem cells, can rapidly repair DNA damage and successfully survive IR.

The protective role of MET likely operates in human GBMs treated with the standard combination of radio- and chemotherapy with the alkylating agent temozolomide (Stupp *et al*, 2009). Indeed, in the patients' cohort analyzed, MET expression is enriched in the recurrent tumors, as compared with the matched primary. This indicates the expansion of MET-expressing (stem-like) cells, likely benefitting of a selective advantage under therapeutic pressure and

**Figure 8. MET inhibition radiosensitizes GSC-derived GBMs by GSC depletion.**

A  Timeline of therapeutic treatments undergone by mice intracranially injected with BT463NS.

B  Left: Representative images of *ex vivo* epifluorescent GFP signal of intracranial tumors, generated by injection of BT463NS, and explanted 62 days after IR in the absence (2 Gy × 3 days) or in the presence (combo) of the MET inhibitor JNJ38877605 (50 mg/kg). Right: Quantification of epifluorescent GFP signal from tumors represented on the left (*n* = 4/condition). Vehicle: non-irradiated vehicle-treated tumors. *: one-way ANOVA, *P* = 5.5 × 10$^{-5}$.

C  Growth curves of tumors, generated by subcutaneous injection of BT308NS, irradiated in the absence (2 Gy × 3 days) or in the presence (combo) of JNJ38877605, which was administered for 15 days as indicated (*n* = 7/condition). Vehicle: non-irradiated vehicle-treated tumors. *: one-way ANOVA, *P* = 0.0006.

D  Survival analysis of mice bearing tumors generated and treated as in (C). Black dot: censored mouse. Log-rank (Mantel-Cox) test (combo) versus (2 Gy × 3 days), *P* = 0.0019.

E  Experimental design to measure GSC frequency in tumors treated as in (C) (JNJ was administered for 10 days). Explanted tumors were dissociated, and viable cells were tested by *in vitro* sphere-forming LDA or *in vivo* single-passage (p1) LDA.

F  *In vitro* LDA (sphere-forming) measuring GSC frequencies as described in (E) (*n* = 3/condition). *: χ$^2$ test, *P* < 10$^{-5}$.

G  Tumor incidence in the *in vivo* LDA performed on tumors generated and treated as described in (E).

H  Histogram showing GSC frequencies measured by *in vivo* LDA, as described in (E). *: χ$^2$ test, *P* = 5.47 × 10$^{-6}$.

I  Table showing GSC frequencies measured by *in vivo* LDA, represented in (H).

Data information: Data are represented as mean ± SEM in (B, C) or ± CI in (F, H, I).

▶

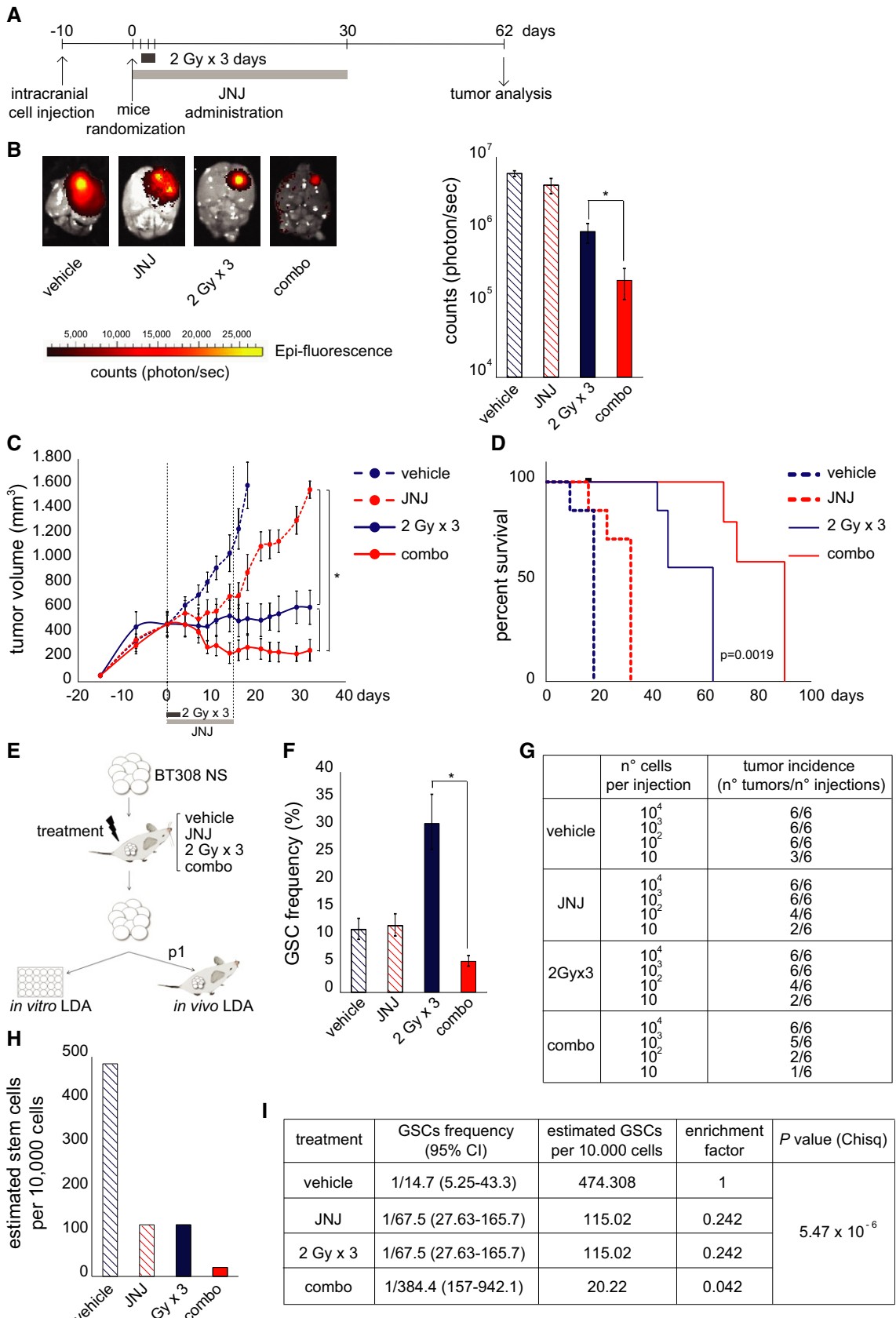

**Figure 8.**

possibly mediating the well-known radiotherapeutic refractoriness of the recurrent tumor. Not surprisingly, analysis of the TCGA dataset indicates that high MET expression in primary GBMs is associated with primary therapeutic resistance (decreased disease-free survival) and poorer prognosis.

This evidence compelled us to investigate the possibility to overcome GSC radioresistance by MET inhibition. Indeed, we show that MET small-molecule kinase inhibitors, or inhibitory antibodies, combined with IR, decrease the radioresistance of GSCs (MET$^{high}$) to levels comparable with non-stem (MET$^{neg}$) cells. We report for the first time that, in irradiated NS, MET inhibition impairs ATM phosphorylation and the ensuing DDR, resulting in dramatic reduction of RAD51 expression, and DNA repair failure. Till now, the relationship between receptor tyrosine kinase signaling and DDR has been poorly characterized. We thus sought to reconstruct the mechanistic link between MET and ATM inhibition, establishing an essential role for AKT. We show that both AKT and MAP kinases, two prominent signal transducers downstream MET, are hyperphosphorylated together with MET in irradiated cells. AKT and MAP kinase hyperactivation depends of MET, as it is prevented by combining IR with MET inhibitors. Moreover, IR combination with either MAPK or AKT inhibitors showed that AKT, but not MAPK, is the likely connection between MET and ATM phosphorylation. Interestingly, AKT activation has been previously correlated with poor prognosis in GBM (Phillips *et al*, 2006a) and with radioresistance in other brain tumors (medulloblastomas) (Hambardzumyan *et al*, 2008). However, it was currently unknown whether AKT can directly affect ATM phosphorylation. We now show that a conceivable intermediary is Aurora kinase A, known to be both an AKT substrate and a kinase active on ATM (Shiloh & Ziv, 2013), but not yet involved in DDR modulation by growth factor signaling.

We also show that AKT activation by MET sustains radioresistance by modulating p21 activity. It is known that, after phosphorylation by AKT, p21 is retained in the cytoplasm and inhibits apoptosis-related transducers and effectors, including procaspases and caspases (Abbas & Dutta, 2009). We show that, after irradiation in the presence of HGF, p21 is phosphorylated and mainly localized in the cytoplasm, whereas, when MET inhibitors are combined with irradiation, p21 phosphorylation by AKT is prevented, and the protein translocates into the nucleus. Accordingly with what observed in this study, p21 nuclear translocation implies loss of anti-apoptotic activity and inhibition of the cell cycle (Abbas & Dutta, 2009). It could be further speculated that, in the presence of MET inhibitors, nuclear p21 prevents CDK1 from stimulating homologous recombination via the BRCA2–RAD51 complex (Esashi *et al*, 2005). Collectively, the above data suggest that MET inhibition radiosensitizes GSCs by blocking AKT activation, which results in (i) decreased ATM activity and (ii) conversion of p21 function from prevention of apoptosis to inhibition of cell cycle and DNA repair.

This study offers preclinical evidence that MET inhibitors, associated with radiotherapy, not only cause tumor shrinkage, but also reduce the content of GSCs, the source of tumor recurrence. Evaluation of the therapeutic effect on cancer stem cells is essential to establish the efficacy of a treatment, but, regretfully, it is often disregarded by the conventional settings testing either radiochemotherapy and/or targeted agents (Baumann *et al*, 2008; Krause *et al*, 2011; Vlashi & Pajonk, 2014).

To target MET in GBM patients, clinically approved MET inhibitors (such as the multi-kinase inhibitor Crizotinib) are available, and new compounds, either small molecules or antibodies, are under development (Peters & Adjei, 2012; Vigna & Comoglio, 2015). Interestingly, beside radiosensitization, MET inhibitors could display therapeutic activity *per se*, interfering with different aspects of GSC involvement in tumorigenesis. Indeed, MET inhibitors can impair the GSC phenotype, sustained through signaling stimulated by the ligand HGF (Li *et al*, 2011; De Bacco *et al*, 2012). Interestingly, HGF is provided by paracrine circuits and often produced in autocrine loops by GBM cells (Xie *et al*, 2012) and NS (Appendix Table S3). Consistently—even alone—MET inhibition can impair tumor growth *in vivo*. Moreover, MET inhibitors could be highly effective in the small percentage of patients (3%) where *MET* is amplified (Brennan *et al*, 2013) and could confer "oncogene addiction" to GSCs (Boccaccio & Comoglio, 2014). Finally, MET inhibition could target neoangiogenic GBM vessels (Michieli *et al*, 2004), thus impairing the perivascular niche, known to be essential for GSC maintenance (Calabrese *et al*, 2007).

Like any other targeted therapy, the success of MET inhibitors or antibodies will ultimately depend on the precise identification of patients expressing the functional target. According to the analysis presented in this study and previous studies (Koochekpour *et al*, 1997; De Bacco *et al*, 2012), it is expected that a relevant fraction of GBM patients (~40%) will be MET positive. In the clinical setting, integration of protein expression and genetic data, obtained by analysis of GBM tissues or more accessible biological samples, will be required to discriminate patients that could benefit from MET targeted therapies.

## Materials and Methods

### NS derivation and culture from human GBM

From surgical samples of consecutive primary GBM (provided by the Fondazione IRCCS Istituto Neurologico C. Besta, according to a protocol approved by the institutional Ethical Committee), NS were derived as described (De Bacco *et al*, 2012) and plated at clonal density in standard medium containing human recombinant bFGF (20 ng/ml) and EGF (20 ng/ml). For *in vitro* experiments, NS were kept in a modified medium with EGF and bFGF (10 ng/ml), supplemented with human recombinant HGF (10 ng/ml), unless otherwise indicated. Both media sustain the same NS proliferation rate (Appendix Fig S9A). NS-derived pseudodifferentiated cells were obtained by culture in medium deprived of growth factors and supplemented with 2% FBS, in pro-adherent culture dishes, for 7 days.

### NS irradiation and treatment

NS were irradiated with a 200 kV X-ray blood irradiator (Gilardoni), and a 1 Gy/min dose rate. In some experiments, HGF (20 ng/ml) was added to standard medium 1 h before irradiation. For MET targeting, the kinase inhibitors JNJ38877605 (500 nM, Janssen Pharmaceutica) (De Bacco *et al*, 2011), PHA665752 (500 nM; Tocris Cookson Ltd), or Crizotinib (500 nM, Sequoia Research Products) were added to cells 2 h before irradiation. Inhibitors for MAPK

**The paper explained**

**Problem**

GBM, the most aggressive and common primary brain tumor, usually remains refractory to the best standard of care, entailing radiotherapy as a mainstay and, often, as the only treatment option. GBM radioresistance has been associated with distinctive properties of the GBM stem-like subpopulation: after irradiation, while bulk cells accumulate DNA damage and die, stem-like cells efficiently activate DNA repair mechanisms and survive, driving tumor recurrence. A deeper understanding of the mechanisms of GBM stem-like cell radioresistance is needed, in order to identify druggable targets for radiosensitization and long-term effective therapeutic response.

**Results**

By analyzing a large panel of GBM stem-like cells *in vitro* propagated as neurospheres, we provide evidence that radioresistance is significantly higher in GBM stem-like cells than in their differentiated counterpart (including cells derived from stem-like cell pseudodifferentiation). We show that the levels of radioresistance are similar in stem-like cells displaying different genetic alterations or transcriptional profiles, which characterize distinct GBM subtypes (classical, proneural, and mesenchymal). However, in a subset of neurospheres, radioresistance is associated with expression of MET, the HGF tyrosine kinase receptor. MET-expressing stem-like cells are positively selected by ionizing radiation *in vitro* and, possibly, also *in vivo*, as assessed in a cohort of human patients including 20 cases of surgically removed primary GBMs and their matched recurrences.

We then provide a mechanistic analysis of the still elusive signaling pathways through which MET promotes radioresistance: one sustains activation of ATM kinase, which orchestrates DNA repair; the other induces p21 cytoplasmic retention, associated with antiapoptotic functions. Finally, *in vitro* and *in vivo*, we show that MET inhibition radiosensitizes tumors, by improving the overall control of tumor growth, and, most importantly, by converting the danger of GBM stem-like cell-positive selection into the benefit of their depletion.

**Impact**

We provide preclinical evidence that MET drugs can be combined with radiotherapy to undermine the inherent radioresistance of GBM stem-like cells, thus increasing the efficacy of radiotherapy in tumor growth control and, possibly, in preventing recurrence.

(PD98059: 20 μM, Calbiochem), AKT (Ly294002: 20 μM, Calbiochem), Aurora kinase A (MLN8237: 20 μM, Millennium Pharmaceuticals), ATM (CGK733: 10 μM, Sigma) were added 1–4 h before IR. DMSO was used as vehicle.

## Xenografts models and treatments

All animal procedures were approved by the Italian Ministry of Health and the internal Ethical Committee for Animal Experimentation. NS-derived single-cell suspensions were injected into 6- to 8-week-old female NOD.CB17-Prkdcscid/NcrCr mice. Subcutaneous tumor diameters were measured every 3 days by caliper. For orthotopic transplantation, cells expressing a bicistronic luciferase-GFP construct were delivered into the right corpus striatum by stereotactic injection. Intracranial tumors were monitored by bioluminescence (BLI) imaging (IVIS® SpectrumCT, Caliper Life Sciences) once a week. For radiosensitization experiments, mice were randomized by LAS software (Baralis *et al*, 2012). IR was delivered by TomoTherapy HD or the equivalent Hi-Art (Accuray, Inc.).

JNJ38877605 (50 mg/kg) was administered by daily oral gavage starting the day before IR. Mice were monitored daily and sacrificed in case of evident suffering. In survival experiments (subcutaneous xenografts), mice were sacrificed when tumor size reached 1,600 mm³; animals euthanized before this endpoint were included in survival curves as censored observations. In the orthotopic model, mice were euthanized at day 62 after start of the treatment (onset of neurological symptoms in the control group). Explanted brains were immediately analyzed for GFP signal detection (e$_x$ 465 nm, e$_m$ 520 nm; IVIS® SpectrumCT).

## Limiting dilution assay *in vivo*

*In vivo* limiting dilution assay (LDA) was performed to evaluate the GSC frequency in subcutaneous xenografts. Tumors were explanted and dissociated to re-derive tumor cells, which were cultured in standard medium as above. After recovery (about 10 days), single-cell suspensions of viable cells (trypan blue exclusion test) were injected in the right flank of mice at the following doses: 10, $10^2$, $10^3$, or $10^4$ cells in 100 μl v/v PBS/Matrigel (six mice/condition). The GSC frequency was evaluated on the efficiency of secondary xenograft formation through the extreme limiting dilution assay (ELDA) software (http://bioinf.wehi.edu.au/software/elda/).

## Statistical analysis

Values were expressed as mean ± standard error of the mean (SEM) or confidence intervals 95 % (CI). *In vitro* experiments were repeated at least twice in quadruplicate. Statistical analyses were performed using GraphPad Prism Software (GraphPad Software Inc.). Unpaired two-sided Student's *t*-test, one-way analysis of variance (ANOVA), Wilcoxon, Mantel–Cox, or chi-squared tests were used as indicated. A *P*-value < 0.05 was considered to be significant.

Detailed methods are available as Appendix Supplementary Methods.

**Expanded View** for this article is available online.

## Acknowledgements

We thank Viola Bigatto for help with some experiments, Livio Trusolino for discussion, Paola Bernabei for cell sorting, Antonella Cignetto, Daniela Gramaglia, and Francesca Natale for assistance, and Guglielmo Castelli for art. This work was supported by Associazione Italiana per la Ricerca sul Cancro ("Special Program Molecular Clinical Oncology 5xMille, N. 9970"; IG 10446 and 15709 to C.B. and N. 11852 to P.M.C.); Ministero della Salute (Ricerca Corrente 2014–2015); and Comitato per Albi98.

## Author contributions

FDB, ADA, and CB designed the experimental strategy. FDB, ADA, EC, FO, RN, RA, FV, and GR planned the experiments and collected and analyzed the data. PL assisted with the experimental methodology. MC and PLP analyzed patient tissue samples. SP and GF obtained neurospheres from patients. TP provided the MET inhibitor JNJ38877605. EG and PG designed protocols for mouse radiotherapy. FDB, PMC, and CB wrote the manuscript. PMC and CB provided financial support. All authors discussed and approved the manuscript.

## Conflict of interest

TP was a former employee of Ortho Biotech Oncology Discovery Research, Janssen Pharmaceutica, that provided the MET inhibitor JNJ38877605, but he was not involved in collection, analysis, and interpretation of results concerning the inhibitor. He is founder of Octimet Oncology Ltd and member of the scientific advisory board. The other authors declare that they have no conflict of interest.

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
