## [Review Process File · EMBO Molecular Medicine]

MET inhibition overcomes radiation resistance of glioblastoma stem-like cells

Francesca De Bacco, Antonio D'Ambrosio, Elena Casanova, Francesca Orzan, Roberta Neggia, Raffaella Albano, Federica Verginelli, Manuela Cominelli, Pietro L. Poliani, Paolo Luraghi, Gigliola Reato, Serena Pellegatta, Gaetano Finocchiaro, Timothy Perera, Elisabetta Garibaldi, Pietro Gabriele, Paolo M. Comoglio and Carla Boccaccio

Corresponding authors: Carla Boccaccio and Paolo M. Comoglio, Candiolo Cancer Institute, FPO-IRCCS

Review timeline:	Submission date:	16 October 2015
	Editorial Decision:	18 November 2015
	Revision received:	05 February 2016
	Editorial Decision:	23 February 2016
	Revision received:	26 February 2016
	Accepted:	02 March 2016

Editor: Roberto Buccione

Transaction Report:

1st Editorial Decision

18 November 2015

Thank you for the submission of your manuscript to EMBO Molecular Medicine. We have now heard back from the three Reviewers whom we asked to evaluate your manuscript. We are sorry that it has taken longer than usual to get back to you on your manuscript. In this case we experienced some difficulties in obtaining the Reviewer evaluations in a timely manner. Further to this, I wished to discuss the evaluations with my colleagues.

As you will some of the issues raised are fundamental and in part shared by the three Reviewers. Although I will not dwell into much detail, I would like to highlight the main points.

Reviewer 1 expresses one main, yet fundamental concern: perceived lack of essential novelty at many levels. This Reviewer also mentions some potential elements of interest, which however require stronger experimental support for consolidation. For instance s/he notes the radiation resistance of some MET-negative neurospheres and would have liked to see some follow-up on this observation. S/he would also like more mechanistic insight into the mechanisms downstream of Met, driving the DDR. This reviewer also lists several other issues that would require attention.

Reviewer 2 is less reserved but does also express similar concerns in terms of insufficient experimental support for some conclusions, including on the link between MET abundance and activation, the fact that high MET CSCs are preferentially killed by the combination treatment and effect of radiations on CSCs.

Reviewer 3 is globally more positive, but expresses a few concerns. The first, similarly to Reviewer 1, is related to the data on radio resistance of the NSs. S/he also notes various instances where the data do not fully support your conclusions.

In conclusion, while publication of the paper cannot be considered at this stage, given the potential interest of your findings, and after discussion, we have decided to give you the opportunity to address the above concerns.

We recognize that some basic concepts are not novel, but appreciate that you use clinically relevant settings to improve on previous work. There is consensus, also at the editorial level, that additional mechanistic insight is needed on the mechanisms driving DDR, which would be novel. We are thus prepared to consider a substantially revised submission, with the appreciate understanding that the Reviewers' concerns must be addressed with additional experimental data where appropriate and that acceptance of the manuscript will entail a second round of review. The overall aim is to significantly upgrade the impact, significance and translational relevance of the dataset, which of course are of paramount importance for our title. I would also encourage you to underscore more clearly the novel aspects of your work. I understand that if you do not have the required data available at least in part, to address the above, this might entail a significant amount of time, additional work and experimentation and might be technically challenging, I would therefore understand if you chose to rather seek publication elsewhere at this stage. Should you do so, and we hope not, we would welcome a message to this effect.

***** Reviewer's comments *****

Referee #1 (Comments on Novelty/Model System):

1. The experiments are well-designed but the questions and conclusions are relatively lacking in novelty and impact for the following reasons. It is well known that GSCs are radiation resistant and that this reflects an upregulation of DDR. It is known that Met marks and drives GSCs and that Met inhibitors deplete cultures and tumor xenografts of radiation resistant GSCs. It is known that Met inhibitors sensitize tumor models including glioma models to radiation and chemotherapy. The DDR molecular pathways implicated in the GSC response to Met inhibitors have for the most part also been previously described. A link between Akt, ATM and DDR has been previously described, including in glioma.

Referee #1 (Remarks):

This manuscript by De Bacco et al examines the role of the Met receptor tyrosine kinase in the radiation resistance of glioblastoma tumor-initiating stem cells (GSCs). The work presented is comprehensive but in many respects derivative in that many of the findings simply extend the considerable amount of already existing knowledge generated by these investigators and others regarding the role of Met as a marker and driver of GSCs, the radiation resistance of GSCs, and the capacity of Met inhibitors to sensitize Met-positive cancer cells and tumor xenograft models including glioblastoma models to DNA damaging agents. The specific experiments are well-described and clearly presented and most but not all conclusions can be support by the results.

Specific Comments.

1. The experiments are well-designed but the questions and conclusions are relatively lacking in novelty and impact for the following reasons. It is well known that GSCs are radiation resistant and that this reflects an upregulation of DDR. It is known that Met marks and drives GSCs and that Met inhibitors deplete cultures and tumor xenografts of radiation resistant GSCs. It is known that Met inhibitors sensitize tumor models including glioma models to radiation and chemotherapy. The DDR molecular pathways implicated in the GSC response to Met inhibitors have for the most part also been previously described. A link between Akt, ATM and DDR has been previously described, including in glioma.

2. The authors convincingly show that Met can regulate GSC radiation response. However, Figure 1A shows that some Met-negative NS are also very radiation resistant, a finding that is given little

attention in this paper. An expanded analysis of these Met-negative NS lines might lead to interesting insights and changes in some of the current conclusions.

3. Many of the conclusions are based only on results obtained from the BS308 NS line. This tends to weaken the generalizability of some conclusions.
4. More details are required to understand how GSC frequencies were calculated and presented in Figs 2D-E and S2B.
5. In Figure 2, it would be very useful to use other markers (e.g. CD-133, Sox2) in addition to PKH-26 to characterize the NS cells that resist radiation.
6. The entire section on Met-expression in recurrent GBMs (page 8, Figs3G-I) is incomplete. The clinical specimens are insufficiently described e.g. the time elapsed between the completion of radiation and the acquisition of recurrent tissue. The finding of increased Met expression at GBM recurrence is c/w the known transition to mesenchymal subtype at GBM recurrence and does not necessarily reflect GSC numbers. This is supported by the IHC showing diffuse expression of elevated Met that may not be limited to the GSC subset.
7. The Annexin V flow analysis for apoptosis shown in Fig 4A is somewhat problematic since the biggest change is the emergence of cells that are both DAPI-pos and Annexin V-pos making it impossible to implicate apoptosis vs other death mechanisms. A time course should be performed in order to be able to conclude if these cells transitioned thru apoptosis or not.
8. There is considerable overlap between the results in Figures 1 and 4. The authors should consider integrating and consolidating these experiments and findings.
9. The section (page 11, Fig 6) focusing on signaling mechanisms downstream of Met that drive DDR (Akt, ATM, p21) is interesting though incomplete. Many of the findings are correlative with experiments limited to the use of a single pharmacologic agent to target a specific intermediary of interest. The p21 observations (top of page 12, Figs 6E) are very preliminary and conclusions speculative. More experiments and controls are needed. In addition, the authors' conclusion that IR induced Met activation is very indirect since effects of total and phosphorylated Met are not examined. The conclusion that MLN8237 mimics the radiosensitizing effect of Met inhibition (Fig 6D) is not convincing (is magnitude change in viability induced by MLN + IR really different from effect of veh + IR?).
10. The in vivo results in Fig. 7 are interesting and anti-tumor effects confirm prior published results showing that Met inhibitors can sensitize GBM xenografts to radiation. The effects on tumor GSC frequency are novel and interesting. More methodological details are needed to understand the methods used for Fig. 6E were obtained. Specifically when exactly were cells isolated relative to the start or completion of IR and JNJ administration? How were differences in input cell viability controlled?
11. Given the apparent requirement for Met to sustain GSCs, is it surprising that in vivo Met inhibition alone had no effect on GSC frequency (Fig 7E)? The authors should show that JNJ actually inhibited Met in vivo. Knowing the effects of in vivo JNJ, IR, and JNJ + IR on tumor cell expression of GSC markers would be very useful toward interpreting these results in the context of the prior in vitro experiments and conclusions.

Referee #2 (Remarks):

The manuscript by De Bacco and colleagues untitled "MET inhibition overcomes radiation resistance of glioblastoma stem-like cells" is an interesting study proposing the protecting role of EMT against radiation in glioblastoma stem cell (GSC) and the consequence in term of putative combination therapy. This manuscript is well written and the experiments convincing for publication in EMBO MM. However, even though the referee is aware that to work with cancer stem cells is always challenging, there are few experiments either that are not convincing or that miss to formally

demonstrate that the effect seen here is directly related to an effect to GSC-CSC or tumor initiating cells.

1) Figure 2: the effect of radiation on CSC appears to be minimal in panel D (0/6 mice versus 2/6) (and then the resulting counting is weird in E). If as shown in Fig.7, the radiation treatment alone is leading to a log drop of "cell/light" that are non CSC (if the model is true), we expect to be able to see at least a 10 time enrichment of tumor initiating cells that should be seen by a condition where there is no mice with tumor in controls treated group and tumor in all (or most) mice in radiotherapy treated group. How is the calculation from D to E possible. May be rather than doing three serial engraftments, the authors should do one serial engraftments but at a more early time after IR treatment to avoid plasticity issues.

2) Figure 5-6: it is not completely clear for the referee what is the link between this high level of MET and the "activity" of MET. Does the authors consider that because MET is up-regulated, this leads to auto-activation. This would need to be shown. Moreover, what is the status of the MET ligand ? is it expressed ? and does MET ligand addition change anything to P21, RAD51 expression, pATM ?

3) Figure 7: The data shown here are of potential great interest. However while the key message of the manuscript is that HighMET CSCs are "killed" by the combo, they are only the panel E of this figure that supports this interpretation and so far this is in vitro (sphere formation). The referee would be much more convinced if the authors perform a serial engraftment: basically treat (alone or combo) the animals with tumors, collect the tumors and then regraft in recipient mice (no need for orthotopic here)... if the authors are right, the recipient mice injected with the tumors collected from the combo should not be able to make new tumors.

Referee #3 (Remarks):

Title: MET inhibition overcomes radiation resistance of glioblastoma stem-like cells

In this paper Bacco et. al. have presented very novel work. This work is very logical progression to come out from Dr. Boccaccio's lab that established importance of MET as glioma stem-like cell marker in their previous work. Experiments are very well designed and carried out to identify MET inhibition as a radio-sensitizer treatment of GBM. And even though Bacco et. al. show very novel findings in this comprehensive work, there are some unexplained observations in the study that needs to be explained before this manuscript can be accepted for publication.

1. Chk2 is a downstream protein to ATM in a signaling cascade. However, in Figure 1E BT308 NS shows that ATM phosphorylation goes up with time after radiation treatment reaching to maximum activity at 6 hrs, while Chk2 phosphorylation is highest at 30 min and almost non-existent at 6 hr when ATM phosphorylation is highest. Authors are requested to explain this confounding data. Similarly in supplementary Figure 1F also ATM and Chk2 phosphorylation seemed to follow independent activation kinetics.

2. In Figure 3 C and F authors demonstrate MET positivity using Flow Cytometer experiment. However, it is very difficult to understand how author estimate % of positive cells. Particularly, histogram for isotype control looks different for every experiment suggesting that %positive cells may be a mere artifact of how %positive cells are calculated due to differences in control after radiation treatment. Authors are requested to repeat the experiment and set the isotype controls to similar levels to truly determine % MET positive cells.

3. Authors claim that Methigh GSC have more efficient DDR as compared to Metneg GSC. However, when we look at Figure 5G (Radiation and Radiation + JNJ treatment of Methigh GSC), it seems that Methigh GSC in presence of active MET signaling do not incur as much DNA damage (γ H2A.X kinetics). Authors are requested to show kinetics of γ H2A.X foci formation in Methigh GSC and Metneg GSC.

4. Figure 6E: Authors are requested to comment on p21 expression at 24 hrs after radiation treatment, as it seems that cytoplasmic p21 disappeared after 24 hrs as compared to non-treated cells. Authors are also requested to provide a better WB for Figure 6E, unclear blot with multiple air bubbles, makes it difficult to interpret the data.

5. Comparing Figure 7D and S7D it is clear that combination therapy (Rad + JNJ) show significant improvement for only one BT model as compared to radiation therapy alone. Authors are requested to comment on this observation if there are any underlying genomic underpinnings for such different behavior for tested models.

1st Revision - authors' response

05 February 2016

POINT-BY-POINT REPLY TO REVIEWERS' COMMENTS

Reviewer #1

Comment on Novelty/Model system

The experiments are well-designed but the questions and conclusions are relatively lacking in novelty and impact for the following reasons. It is well known that GSCs are radiation resistant and that this reflects an upregulation of DDR. It is known that Met marks and drives GSCs and that Met inhibitors deplete cultures and tumor xenografts of radiation resistant GSCs. It is known that Met inhibitors sensitize tumor models including glioma models to radiation and chemotherapy. The DDR molecular pathways implicated in the GSC response to Met inhibitors have for the most part also been previously described. A link between Akt, ATM and DDR has been previously described, including in glioma.

Reply

We thank the Reviewer for her/his appreciation of the experimental design. Concerning novelty, we agree with this Reviewer that the ability of MET inhibitors to radiosensitize tumor xenografts has been previously shown, by us and others. This was an essential prerequisite to further investigate MET inhibition in glioblastoma stem-like cells. We honestly think that there was no previous convincing evidence that “*Met inhibitors deplete cultures and tumor xenografts of radiation resistant GSCs*”: this lack of knowledge motivated us to tackle the difficult and time-consuming enterprise of testing the effect of radiotherapy + MET inhibitors in glioblastoma stem-like cells *in vitro* and *in vivo*, hardly a matter for easy confirmatory science. On the contrary, as recognized also by the two other Reviewers, this is novel and challenging work. As we are convinced, by evidence obtained by us and others, that glioblastoma stem-like cells are the major culprit of glioblastoma radioresistance, we thought it was essential to show systematically and in details the contribute of MET, with the ultimate purpose to provide significant preclinical data to support clinical trials. In the revised manuscript we added further experiments to show that radiotherapy + MET inhibitors abate glioblastoma stem-like cell frequency *in vivo*, a finding defined by this Reviewer as “novel and interesting” (please see Reply to Specific Comments N. 10 and 11).

Likewise, we could not find that the “*DDR molecular pathways implicated in the GSC response to Met inhibitors have for the most part also been previously described*”. Thus, we set out to accomplish also this task, which is particularly laborious in neurospheres. We fully agree that this issue was only in part resolved in the first version of the manuscript. Thus, now we did our best to adequately strengthen the identification of the molecular mechanisms linking MET to the DDR response, namely (i) the AKT-Aurora kinase A-ATM pathway (where Aurora kinase has never been described before as an intermediary); (ii) the AKT-p21 pathway, whose modulation by MET is, as far as we know, a full novelty (please see reply to Specific Comment N. 9 for a detailed list of new results).

Specific comment N. 1

The experiments are well-designed but the questions and conclusions are relatively lacking in novelty (...)

Reply

As this comment fully overlaps with the above “*Comment on Novelty/Model system*”, please see the related reply.

Specific comment N. 2

The authors convincingly show that Met can regulate GSC radiation response. However, Figure 1A shows that some Met-negative NS are also very radiation resistant, a finding that is given little attention in this paper. An expanded analysis of these Met-negative NS lines might lead to interesting insights and changes in some of the current conclusions.

Reply

In the first paragraph of the Results we showed that radioresistance is a common property of neurospheres (i.e. cultures enriched in stem/progenitor cells), as compared with cells with differentiated features. As shown in Supplementary Fig. S1A,C,D of the original version, now Expanded View 1A-C, neurosphere radioresistance could not be preferentially associated with any genetic alteration, or gene expression profile, and thus was not specific of MET-expressing neurospheres (please see Results, Page 5 of both the original and revised version). The study was then focused on MET-positive neurospheres (about 50% of more than 100 analyzed) as the MET-negative mostly harbor EGFR amplification (and display a classical gene expression profile), as recalled in the Introduction (Page 3, 2nd paragraph), referring to previous studies (De Bacco et al., Cancer Res. 72:4537). In MET-negative glioblastoma stem-like cells, thus, it is conceivable that EGFR plays a prominent role in driving radioresistance, as suggested by other authors (Squatrito and Holland, Cancer Res. 71:5945). We are currently investigating this issue, which implies a considerable experimental effort, parallel to MET studies. As EGFR and MET seem to have similar roles, but are alternatively associated with different neurosphere subtypes (e.g. MET-mesenchymal vs. EGFR-classical), we frankly believe that analysis of EGFR in neurosphere radioresistance goes beyond the scope of this paper. Taking in consideration the Reviewer's concerns, a further comment on neurosphere radioresistance, which can rely also on mechanisms different from MET in MET-negative stem-like cells, has been added to the Discussion (Page 15, First paragraph). To further avoid misunderstandings, (i) in the Abstract we now specify that MET is expressed "in a subset of radioresistant GSCs"; (ii) in Results (Page 7) we reiterate that we investigated the relationship between MET expression and GSC radioresistance "in a subset of NS"; (iii) in Paper Explained (Page 21) we specify that "in a subset of neurospheres, radioresistance is associated with expression of MET".

Specific comment N. 3

Many of the conclusions are based only on results obtained from the BS308 NS line. This tends to weaken the generalizability of some conclusions.

Reply

Please note that, already in the original version, for the majority of experiments, results on BT308 neurosphere were usually shown in the main Figures, while overlapping results obtained with at least another neurosphere, and in many cases with 3 other neurospheres, were represented in Supplementary Figures. *In vivo* experiments testing the ability of MET inhibitors to radiosensitize glioblastoma stem cells were performed on BT463 (intracranial model), and on BT308 and BT371 (subcutaneous models).

To further strengthen the generalizability of the conclusions, in the revised version we repeated several crucial experiments with additional neurospheres. Among the others, please note:

- (i) Concerning radioresistance of MET^{high} and MET^{neg} sorted subpopulations, in new Supplementary Fig. S4, in panel B and C the surviving fraction after irradiation is shown in 5 other neurospheres; in panel E comet assay is shown also in BT205; in panel F ATM and Chk2 activation are also shown in BT452.
- (ii) Concerning studies on radiotherapy + HGF or MET inhibitors *in vitro*, in new Supplementary Fig. S5, in panel A and B, viability in the presence of HGF is also shown in BT463 and BT205; in panel E and F, viability and caspase-3 activation are shown in BT463; in panel I, the surviving fraction is also shown in sorted BT463.
- (iii) Concerning studies on DDR activation by MET, which has been amply revised, experiments were performed in at least two neurospheres (new Fig. 6 and 7, and Supplementary Fig. S6 and S7; please see Reply to Specific Comment N. 9 for details).

Specific comment N. 4

More details are required to understand how GSC frequencies were calculated and presented in Figs 2D-E and S2B.

Reply

More details, concerning GSC frequency calculation by *in vivo* limiting dilution assays, represented in Fig. 2D-E (now Fig. 2E-F) and in the new Fig. 8G-I, have been added in the Methods section. This section has been moved from Supplementary Information to the main text (Page 19). Moreover (i) the output of the ELDA software, used for GSC calculation, previously shown in former Supplementary Fig. S2B, has been included in the Legend to Fig. 2F; (ii) for the new *in vivo* LDA on treated tumors (please see Reply to Specific Comment N. 11 for details) raw data for GSC calculation and the output of the ELDA software have been shown in new Fig. 8G,I .

Specific comment N. 5

In Figure 2, it would be very useful to use other markers (e.g. CD-133, Sox2) in addition to PKH-26 to characterize the NS cells that resist radiation.

Reply

In the new Fig. 2B, and new Supplementary Fig. S2B, we added SOX2 transcription measurement by qPCR, and Olig2 detection by flow-cytometry, in several neurospheres, to better characterize radioresistant cells. As mentioned in the Result section (Page 6), Olig2 was chosen as it was previously used to characterize mesenchymal/proneural radioresistant neurospheres (Bhat et al. Cancer Cell 24:331). CD133 was not used as it is poorly expressed in MET-positive neurospheres (De Bacco et al., Cancer Res. 72:4537; Lottaz et al. Cancer Res. 70:2030).

Specific comment N. 6

The entire section on Met-expression in recurrent GBMs (page 8, Figs3G-I) is incomplete. The clinical specimens are insufficiently described e.g. the time elapsed between the completion of radiation and the acquisition of recurrent tissue. The finding of increased Met expression at GBM recurrence is c/w the known transition to mesenchymal subtype at GBM recurrence and does not necessarily reflect GSC numbers. This is supported by the IHC showing diffuse expression of elevated Met that may not be limited to the GSC subset.

Reply

The description of the clinical specimens has been enriched with details on clinical data, including, as requested, the time elapsed between the completion of radiation and the acquisition of recurrent tissue (new Supplementary Table S4). Details on therapy were added also in the Supplementary Methods section (Page 32).

Concerning the comment that “*the finding of increased Met expression at GBM recurrence is c/w the known transition to mesenchymal subtype at GBM recurrence and does not necessarily reflect GSC numbers*”, we agree that increased expression of MET (a mesenchymal marker) reflects the already known transition to mesenchymal subtype in recurrence. However, we’d like to point out that our findings on MET radioresistance, for the first time, indicate that MET can play a functional role in driving this mesenchymal transition of the recurrence.

Moreover, we agree that we do not provide a functional proof that, in patients, the MET-positive cells increased in recurrences are all stem-like cells. Being aware of this limitation, in the original version of the manuscript we entitled the Result section “MET-expressing (stem-like) cells are enriched in recurrent human GBMs”, and used the word “stem-like” in brackets throughout the section, to express caution in drawing such conclusions. However, we feel it is reasonable to postulate that the MET-positive cell population expanded in recurrence could correspond to an expanded stem-like population, based on two pieces of evidence: (i) in MET-pos neurospheres, the MET-negative cell subpopulation is devoid of stem properties, as shown in previous work (De Bacco et al., Cancer Res. 72:4537), and further investigated by the stem cell frequency assay (limiting dilution assay) shown in Fig. 3B and Supplementary Fig. S3A; (ii) GSC differentiation is characterized by loss of MET expression, likely due to accumulation of miRNAs (miR34a and 23b) targeting MET, as previously shown (De Bacco et al., Cancer Res. 72:4537). Finally, it should be also considered that GBM recurrences seem enriched in stem-like cells, as they are more aggressive, and more prone to generate neurospheres than primary GBM (our unpublished data). Nevertheless, to avoid overstatement, we deleted the word “stem-like” from the title and throughout the Result paragraph, and we added a statement to justify our hypothesis about the stem-like identity of the expanded MET-expressing cell population (Page 8).

Specific comment N. 7

The Annexin V flow analysis for apoptosis shown in Fig 4A is somewhat problematic since the biggest change is the emergence of cells that are both DAPI-pos and Annexin V-pos making it

impossible to implicate apoptosis vs other death mechanisms. A time course should be performed in order to be able to conclude if these cells transitioned thru apoptosis or not.

Reply

We agree with the Reviewer that in Fig. 4A it was impossible to discriminate whether cell death was due to apoptosis or other death mechanisms. We appreciate the suggestion to show a time-course Annexin V analysis. However, we think that apoptosis of MET^{neg} cells, sorted and irradiated, is better shown, and enriched with a mechanistic detail, by time-course Western Blots showing activation of PARP and caspase-3, well-known apoptosis markers (new Fig. 4B and Results, Page 9).

Specific comment N. 8

There is considerable overlap between the results in Figures 1 and 4. The authors should consider integrating and consolidating these experiments and findings.

Reply

We thoroughly considered this suggestion, with the aim to avoid unnecessary overlapping while keeping the maximum clarity. We think that this merge would generate confusion, as the first part of the paper (Fig. 1 and 2, Expanded View 1 and 2, and Supplementary Fig. S1 and 2) is devoted to show general radioresistance of neurospheres and stem-like cells, regardless of any specific molecular/genetic features. Results shown in Fig. 4 specifically concern MET-positive NS, and compare the MET^{high} and MET^{neg} cell subpopulations contained in MET-positive NS. It seems that combining these experiments with those of Fig. 1 would result in a puzzling anticipation of data related to MET expression, and overloaded information in the first part of the paper.

Specific comment N. 9

The section (page 11, Fig 6) focusing on signaling mechanisms downstream of Met that drive DDR (Akt, ATM, p21) is interesting though incomplete. Many of the findings are correlative with experiments limited to the use of a single pharmacologic agent to target a specific intermediary of interest. The p21 observations (top of page 12, Figs 6E) are very preliminary and conclusions speculative. More experiments and controls are needed. In addition, the authors' conclusion that IR induced Met activation is very indirect since effects of total and phosphorylated Met are not examined. The conclusion that MLN8237 mimics the radiosensitizing effect of Met inhibition (Fig 6D) is not convincing (is magnitude change in viability induced by MLN + IR really different from effect of veh + IR?).

Reply

The entire study of DDR modulation by MET has been revised, according to welcome suggestions by all three Reviewers. Briefly, we added completely new sets of experiments (i) showing MET hyperphosphorylation by irradiation; (ii) comparing DDR activation in neurospheres irradiated in the presence or the absence of the MET ligand HGF; (iii) clarifying the p21 response to MET activation/inhibition in irradiated cells. Moreover, we added new experiments with more inhibitors (pharmacologic agents) of MET or various members of the DDR pathway, to strengthen the ability of the MET/AKT/Aurora kinase A pathway to sustain ATM activation. Therefore former Fig. 6 and Supplementary Fig. S6 have been extended and split (new EV4, Fig. 6, Supplementary Fig. S6, Fig. 7, and Supplementary Fig. S7). In details:

- 1- We added experiments showing that IR not only induces MET expression (former Supplementary Fig. 3C, now EV3A), but also MET hyperphosphorylation (Page 10, new EV4A,B).
- 2- We show that, in irradiated MET-pos-NS, HGF increases ATM and Chk2 phosphorylation, and RAD51 expression (Page 11, new Fig. 6A and Supplementary Fig. S6A).
- 3- ATM inhibition (new Fig. 6B,C) has been investigated with an additional MET inhibitor (PHA665752; new Supplementary Fig. S6B,C).
- 4- Identification of Aurora Kinase A as the putative intermediary between AKT and ATM (Page 12) has been corroborated by showing that: (i) Aurora kinase A phosphorylation on Thr288, required for enzymatic activation, is prevented by the AKT inhibitor (new Fig. 6E and Supplementary Fig. S6H), as well as by the MET inhibitors JNJ38877605 or PHA665752 (new Fig. 6F and Supplementary Fig. S6I); (ii) Aurora kinase A phosphorylation was increased by HGF (new Fig. 6G and Supplementary Fig. S6J); (iii) Aurora kinase A inhibition by its specific

- small molecule inhibitor MLN8237 prevented ATM phosphorylation (new Fig. 6H and Supplementary Fig. S6K).
- 5- Concerning former Fig. 6D (new Fig. 6I), we confirmed radiosensitization by the Aurora Kinase inhibitor MLN8237 and the AKT inhibitor Ly294002 in two additional neurospheres (new Supplementary Fig. S6L,M). Concerning the observation of this Reviewer about the significance of radiosensitization by MLN, we confirm that MLN significantly reduces viability as compared to ctrl non-irradiated cells (red vs. blue column), although this reduction is minor than with JNJ. This is expected, as MLN is known to reduce viability also in the absence of radiation. Please note that MLN and JNJ reduce viability to a similar extent, with respect to radiation alone (veh, red column).
 - 6- Concerning p21 (Page 12), we now show that, in irradiated MET-pos-NS kept in the standard EGF/bFGF medium, addition of HGF induces p21 phosphorylation on Thr145 (Fig. 7A), and its cytoplasmic accumulation (Fig. 7A). In a complementary experiment, in irradiated MET-pos-NS kept in the medium supplied with HGF, addition of the MET inhibitor prevents p21 phosphorylation on Thr145 (Fig. 7B, Supplementary Fig. S7A), and induced its nuclear translocation (Fig. 7C-E, Supplementary Fig S7B,C). To investigate p21 localization, in addition to Western blots we now show also immunofluorescence experiments.

Specific comment N. 10

The effects on tumor GSC frequency are novel and interesting. More methodological details are needed to understand the methods used for Fig. 6E were obtained. Specifically when exactly were cells isolated relative to the start or completion of IR and JNJ administration? How were differences in input cell viability controlled?

Reply

We thank the Reviewer for his/her appreciation of this part of the work. Concerning methodological details of experiments represented in Fig. 7E, now Fig. 8F, the timing of cell isolation after start of the treatment (10 days, when tumor regression is observed) has been specified in the Result section (Page 13). The cell viability control (trypan blue exclusion of dead cells), used for *in vitro* and *in vivo* limiting dilution assays, has been specified in Methods section (Page 19) and Supplementary Methods (Page 29)

Specific comment N. 11

*Given the apparent requirement for Met to sustain GSCs, is it surprising that in vivo Met inhibition alone had no effect on GSC frequency (Fig 7E)? The authors should show that JNJ actually inhibited Met in vivo. Knowing the effects of in vivo JNJ, IR, and JNJ + IR on tumor cell expression of GSC markers would be very useful toward interpreting these results in the context of the prior *in vitro* experiments and conclusions.*

Reply

The fact that *in vivo* MET inhibition has no effect on GSC frequency is not surprising, if we consider that JNJ likely impairs self-renewal without killing stem-like cells. This is indicated by *in vitro* experiments showing that (i) the clonogenic (sphere forming) ability of MET^{high} cells is impaired when evaluated in the continuous presence of JNJ (former Fig. 5F, now Fig. 5H), but (ii) JNJ alone does not induce stem-like cell apoptosis (former Fig. 5C and Supplementary Fig. S5F, now Fig. 5E and Supplementary Fig. S5F,H). Moreover, please also consider that MET is not an absolute requirement to sustain GSCs. Indeed, other factors such as FGF, present in the tumor microenvironment, contribute to GSC propagation (De Bacco et al., Cancer Res. 72:4537).

Concerning the demonstration that JNJ actually inhibits MET *in vivo*, in the original version of the manuscript we showed not only that JNJ inhibits MET *in vivo*, but also that it crosses the brain-blood barrier (Page 13, former Supplementary Fig. S7A, now Supplementary Fig. S8A). Previous work has shown JNJ efficacy in subcutaneous xenografts of various origin (e.g. Bardelli et al., Cancer Discovery 3:658; De Bacco et al., JNCI 103:645)

Finally, we agree that the suggested analysis of GSC marker expression after *in vivo* treatments would be useful to better investigate whether GSCs are destroyed by the combination of MET inhibitors with radiotherapy. However, we sought to provide a more rigorous, functional demonstration of GSC depletion by adding a crucial experiment, still ongoing at the time of the original submission: GSC frequency evaluation after tumor therapy by *in vivo* limiting dilution assay (LDA) (Fig. 8G-I, Page 13).

Please note that, in LDA performed *in vivo*, the GSC frequency in tumors treated with JNJ alone is decreased (as it would be expected by this Reviewer), while, as discussed above, the GSC frequency measured by *in vitro* LDA was unchanged as compared with control tumors. This could be explained by the fact that the *in vitro* and *in vivo* LDA experimental settings can influence the viability of GSCs, *in vivo* treated with JNJ, in differing ways. In *in vitro* LDA, the presence of growth factors can promote recovery of GSCs, which are inhibited but not killed by previous JNJ treatment (see above). On the contrary, in *in vivo* LDA, the relative lack of growth factors can further compromise GSC viability. In particular, it is known that murine HGF does not cross-react with human MET expressed by GSCs (Luraghi et al. Cancer Res. 74:1857, 2014; Zhang et al. Oncogene 24:101, 2005).

Reviewer #2

General comment

The manuscript by De Bacco and colleagues untitled "MET inhibition overcomes radiation resistance of glioblastoma stem-like cells" is an interesting study proposing the protecting role of MET against radiation in glioblastoma stem cell (GSC) and the consequence in term of putative combination therapy. This manuscript is well written and the experiments convincing for publication in EMBO MM. However, even though the referee is aware that to work with cancer stem cells is always challenging, there are few experiments either that are not convincing or that miss to formally demonstrate that the effect seen here is directly related to an effect to GSC-CSC or tumor initiating cells.

Reply

We thank this Reviewer for his/her kind appreciation of the experimental design, the challenge of working with cancer stem cells, and the potential therapeutic implications of the findings.

Specific comment N. 1

Figure 2: the effect of radiation on CSC appears to be minimal in panel D (0/6 mice versus 2/6) (and then the resulting counting is weird in E). If as shown in Fig.7, the radiation treatment alone is leading to a log drop of "cell/light" that are non CSC (if the model is true), we expect to be able to see at least a 10 time enrichment of tumor initiating cells that should be seen by a condition where there is no mice with tumor in controls treated group and tumor in all (or most) mice in radiotherapy treated group. How is the calculation from D to E possible. May be rather than doing three serial engraftments, the authors should do one serial engraftments but at a more early time after IR treatment to avoid plasticity issues.

Reply

According to the Reviewer expectation, by the *in vivo* limiting dilution assay represented in Fig. 2 C-E of the original version, now Fig. 2D-F, we indeed evaluate a stem cell frequency increased by 11 times, in tumors originated by neurospheres treated with irradiation (NS-IR), vs. untreated neurospheres (NS-ctrl). This is shown in Fig. 2E of the original version, now Fig. 2F. We think that the Reviewer is surprised that this one-log difference in stem cell frequency has been connected with seemingly minimal differences in tumor incidence in mice (0/6 vs. 2/6 mice at 10 injected cells), reported in Fig. 2D of the original version, now Fig. 2E. We think the estimated stem cell frequency is fully reliable, as it has been calculated by the use of the standard, openly available, Extreme Limiting Dilution Assay (ELDA) software (<http://bioinf.wehi.edu.au/software/elda/>), as described in the Method section "Limiting dilution assay *in vivo*", now moved to the main text (Page 19). Please note that evaluation of stem cell frequency (by the software) is obtained by considering tumor incidence at every dilution used (10^4 -10), and that different tumor incidence between the two groups is observed at 10^3 , 10^2 , and 10 dilution (former Fig. 2D, now Fig. 2E). As to openly answer the question "How is the calculation from D to E possible?" and render the data fully transparent, we moved the output of ELDA from former Supplementary Fig. S2B to Fig.2F legend.

Specific comment N.2

Figure 5-6: it is not completely clear for the referee what is the link between this high level of MET and the "activity" of MET. Does the authors consider that because MET is up-regulated, this leads

to auto-activation. This would need to be shown. Moreover, what is the status of the MET ligand? is it expressed? and does MET ligand addition change anything to P21, RAD51 expression, pATM?

Reply

Concerning the link between MET expression and activity, we now show in Fig. EV4 that irradiation of MET-pos-NS induces MET hyperphosphorylation. Please note that, as already described in the original version (Methods section), and now further clarified in the Results section (Page 10), neurospheres are kept in a standard EGF/bFGF medium supplied with HGF, which induces basal MET activation. Moreover, as noticed by the Reviewer, the increase in MET phosphorylation induced by irradiation associates with MET upregulation, shown in former Supplementary Fig. S3C-F, now Fig. EV3, accordingly to previous results in cell lines (De Bacco et al., JNCI 103:645).

Concerning the status of the MET ligand HGF, please note that in Supplementary Table S3 we already showed the expression levels of HGF mRNA and protein in neurospheres. These levels are generally modest (2-5 fold lower if compared with cells known to be HGF producers, such as fibroblasts or astrocytes; data not shown). Moreover, in neurospheres we could not detect a significantly increased HGF expression after irradiation, unlike previously shown in fibroblasts (De Bacco et al., JNCI 103:645). We concluded that MET activation after irradiation does not rely on increased autocrine HGF. However, we think that exogenous HGF is critical for the experimental setting. Indeed, as stated in the Results (Page 10) and Discussion sections (Page 17), HGF is ubiquitously present in the brain (tumor) tissue, where the main source should be glial/microglial cells and endothelia (Kunkel et al., Neuro Oncol 3:82, 2001).

The question concerning the effect of HGF stimulation on DDR effectors has been investigated with new experiments, showing that, after adding *ex-novo* HGF to neurospheres kept in the standard EGF/bFGF medium, ATM, Chk2, Aurora Kinase A, and p21 get phosphorylated, and RAD51 is overexpressed. These results have been reported in: (i) Page 11, and new Fig. 6A,G and Supplementary Fig. S6A,J (ATM, Chk2, RAD51 and Aurora A); (ii) Page 12, and new Fig. 7A (p21).

Specific comment N.3

Figure 7: The data shown here are of potential great interest. However while the key message of the manuscript is that HighMET CSCs are "killed" by the combo, they are only the panel E of this figure that supports this interpretation and so far this is in vitro (sphere formation). The referee would be much more convinced if the authors perform a serial engraftment: basically treat (alone or combo) the animals with tumors, collect the tumors and then regraft in recipient mice (no need for orthotopic here)... if the authors are right, the recipient mice injected with the tumors collected from the combo should not be able to make new tumors.

Reply

We thank this Reviewer for the appreciation of these data, and we fully agree that the suggested experiment, a serial transplantation/limiting dilution assay *in vivo*, would be essential to convincingly show that the combination of radiotherapy and MET inhibitors can effectively kill stem-like cells. This laborious experiment, initiated before the submission of the first version of the manuscript, was successfully concluded, and is now reported in new Fig. 8E,G-I and Supplementary Fig. S8E,F (Page 13).

Reviewer #3

General comment

In this paper Bacco et. al. have presented very novel work. This work is very logical progression to come out from Dr. Boccaccio's lab that established importance of MET as glioma stem-like cell marker in their previous work. Experiments are very well designed and carried out to identify MET inhibition as a radio-sensitizer treatment of GBM. And even though Bacco et. al. show very novel findings in this comprehensive work, there are some unexplained observations in the study that needs to be explained before this manuscript can be accepted for publication.

Reply

We thank this Reviewer for his/her kind praise of the study, and for emphasizing that it is "logical progression" of previous work, and it is "very novel" and "comprehensive". We think he/she alludes

to the fact that radiosensitization of glioma stem-like cells by MET inhibitors has never been formally and thoroughly proven before.

Specific comment N.1

Chk2 is a downstream protein to ATM in a signaling cascade. However, in Figure 1E BT308 NS shows that ATM phosphorylation goes up with time after radiation treatment reaching to maximum activity at 6 hrs, while Chk2 phosphorylation is highest at 30 min and almost non-existent at 6 hr when ATM phosphorylation is highest. Authors are requested to explain this confounding data. Similarly in supplementary Figure 1F also ATM and Chk2 phosphorylation seemed to follow independent activation kinetics.

Reply

We understand the concern of this Reviewer about the different phosphorylation kinetics between ATM and its downstream effector Chk2. However, we think that these differences are not in conflict with dependence of Chk2 phosphorylation of ATM in our experimental setting. To our knowledge, the phosphorylation kinetic of a downstream kinase can be very different from that of its upstream kinase. In particular, negative feed-backs (e.g. specific phosphatases) can specifically act on the downstream kinase, reducing phosphorylation even when the upstream kinase is still (or even more) active: this would explain why Chk2 is not phosphorylated when ATM reaches its peak phosphorylation. We should also consider that ATM functions (connected with ATM phosphorylated status) extend well beyond Chk2 activation (Shiloh and Ziv, Nat. Rev. Mol. Cell Biol. 14:197).

Specific comment N. 2

In Figure 3 C and F authors demonstrate MET positivity using Flow Cytometer experiment. However, it is very difficult to understand how author estimate % of positive cells. Particularly, histogram for isotype control looks different for every experiment suggesting that %positive cells may be a mere artifact of how %positive cells are calculated due to differences in control after radiation treatment. Authors are requested to repeat the experiment and set the isotype controls to similar levels to truly determine % MET positive cells.

Reply

Concerning Fig. 3C (please note that Fig. 3F does not show flow cytometry, but a morphometric quantification of Fig. 3E), we are sorry that representation of MET positive cells by flow-cytometry raised methodological doubts about the isotype control, probably due to insufficient description in the Methods section. “Isotype control” is performed by analyzing a cell sample not stained with MET (or any other) antibodies for every experimental condition (ctrl, 2Gy×3, and 5Gy), as to take in account morphological, and possible ensuing autofluorescent, changes associated with treatments. Therefore, this analysis, as noticed by this Reviewer, generates different grey histograms, required for setting the threshold to reliably evaluate the percentage of MET-positive cells in each experimental condition. This procedure, overcoming the use of isotype control, is encouraged (Maecker and Trotter, Cytometry Part A 69A:1037, 2006). Based on these considerations, we think that, in irradiated cells, “setting the isotype controls to similar levels to truly determine % MET positive cells” would lead to artifactual evaluation. To better clarify the histograms (i) we added methodological details for negative control in the Supplementary Methods (Page 28); and (ii) in new Fig. 3C and Supplementary Fig. S3B we added a dotted line to indicate the histogram area selected for calculation of % positive cells (please note that, in spite of differences, all controls define a 10^1 threshold for positivity).

Specific comment N.3

Authors claim that Methigh GSC have more efficient DDR as compared to Metneg GSC. However, when we look at Figure 5G (Radiation and Radiation + JNJ treatment of Methigh GSC), it seems that Methigh GSC in presence of active MET signaling do not incur as much DNA damage γ H2A.X kinetics). Authors are requested to show kinetics of γ H2A.X foci formation in Methigh GSC and Metneg GSC.

Reply

We are sorry that the concise description (text and legend) of the experiment shown in former Fig. 5G, now Fig. 5I, likely caused a misunderstanding. In this experiment (as well as in former Supplementary Fig. S5G, now Supplementary Fig. S5J) we evaluated the effect of radiation and

radiation + JNJ treatment on whole MET-pos-NS, and not in MET^{high} GSCs, as apparently got by this Reviewer. We further clarified this point in the main text (Page 11) and legend to Fig. 5I.

Concerning the interpretation of the experiment, as expected, in the presence of active MET signaling (i.e. treatment with radiotherapy alone, blue line), neurospheres have less phosphorylated H2AX (i.e. less DNA damage) than after treatment with radiotherapy + MET inhibitor (red line). Concerning the request of showing the effect of MET inhibition on H2AX phosphorylation kinetic in sorted MET^{high} and MET^{neg} subpopulations, we think that this experiment is challenging, due to the high number of cells requested for the time-course analysis. Taking into consideration the above clarification, and the other experiments showing that the MET inhibitor radiosensitizes MET^{high} cells (former Fig. 5E,F, now Fig. 5G,H), we wonder whether further analysis of γ H2AX is essential.

Specific comment N. 4

Figure 6E: Authors are requested to comment on p21 expression at 24 hrs after radiation treatment, as it seems that cytoplasmic p21 disappeared after 24 hrs as compared to non-treated cells. Authors are also requested to provide a better WB for Figure 6E, unclear blot with multiple air bubbles, makes it difficult to interpret the data.

Reply

As suggested by this Reviewer and Reviewer #1, to better investigate the signals linking MET to DDR, we fully revised the results presented in former Fig. 6. A detailed study of p21 expression/phosphorylation after irradiation and MET activation/inhibition (including new Western Blots and immunofluoresces) is now presented in new Fig. 7 and Supplementary Fig. S7 (Page 12). For a detailed list of changes, please see reply to Reviewer #1, specific comment N.9. We hope we have now convincingly shown that radiation + JNJ prevents p21 phosphorylation and promotes its translocation to the nucleus.

Specific comment N. 5

Comparing Figure 7D and S7D it is clear that combination therapy (Rad + JNJ) show significant improvement for only one BT model as compared to radiation therapy alone. Authors are requested to comment on this observation if there are any underlying genomic underpinnings for such different behavior for tested models.

Reply

We thank the Reviewer for the suggestion to investigate the possible genomic factors underlying the different response of the two neurospheres transplanted subcutis (BT308 and BT371) to combination treatment. Please note that in the subcutis model BT308 the combination therapy (rad + JNJ) induces clinical regression (> 50% tumor volume reduction, former Fig. 7C-D, now Fig. 8C-D and Supplementary Fig. S8E). In addition, the combination therapy induces a highly significant reduction in epifluorescent signal, suggestive of a proportional tumor volume reduction, in the BT463 orthotopic model (former Fig. 7B, now Fig. 8B). In the subcutis model BT371 (former Supplementary Fig. S7C-D, now Fig. S8C-D), the effect of combination therapy on tumor volume reduction is statistically significant ($P < 0.05$), although it cannot be defined as regression, but only stabilization (now Supplementary Fig. S8C, Page 13). As noticed by this Reviewer, the therapeutic effect on BT371 is associated with a survival improvement (Supplementary Fig. S7D, now Fig. S8D) less relevant than that observed for BT308, although still statistically significant (Mantel-Cox test: $p < 0.0001$). It would be interesting to investigate the concurrent factors that modify the response of different neurospheres to combination treatment, as to provide further criteria for patient stratification. However, the genetic analysis of neurospheres (Supplementary Table S2) does not indicate any significant difference between BT308 and BT371. To explain the different response, we observed that BT371 neurospheres (i) are among the most radioresistant neurospheres (see Fig. EV1 and Fig. 1); (ii) express uncommonly high levels of MET in at least 90% cells (Supplementary Table S3), which could be difficult to fully inhibit *in vivo* and possibly explains high radioresistance; (iii) generate aggressive tumors with large necrotic areas, which can limit uptake of the inhibitor. To better acknowledge the different therapeutic response of BT371 tumors as compared with BT308, we modified the text as follows: “A significant growth inhibition (~ 2-fold) by combination therapy as compared with radiotherapy alone was observed also in tumors generated by transplantation of BT371 NS, although these tumors were not arrested by any therapy (Supplementary Fig S8B-D)” (Page 13).

Thank you for the submission of your revised manuscript to EMBO Molecular Medicine. We have now received the enclosed reports from the referees that were asked to re-assess it.

As you will see your revision was exceptionally well received by the reviewers, who are now globally supportive. I am thus pleased to inform you that we will be able to accept your manuscript pending the following final amendments:

- 1) Please note Reviewer 3's requests to include loading controls for some western blots. S/he also suggests showing the full membrane for Fig. 4B. Finally, the reviewer would like you to provide a few more detail on various parameters in the legends.
- 2) We are now encouraging the publication of source data, particularly for electrophoretic gels and blots, with the aim of making primary data more accessible and transparent to the reader. Would you be willing to provide a PDF file per figure that contains the original, uncropped and unprocessed scans of all or at least the key gels used in the manuscript? The PDF files should be labeled with the appropriate figure/panel number, and should have molecular weight markers; further annotation may be useful but is not essential. The PDF files will be published online with the article as supplementary "Source Data" files. If you have any questions regarding this just contact me. You might take the opportunity to provide fig 4 as source data instead of changing the original figure as suggested by Reviewer 3.

I look forward to reading a new revised version of your manuscript as soon as possible.

***** Reviewer's comments *****

Referee #1 (Comments on Novelty/Model System):

This revised manuscript is substantially strengthened. The novelty is still felt to be somewhat lower than "high" due to issues raised in the initial review. However, novelty is improved by new and stronger data regarding the roles of aurora kinase, ATM and p21 as downstream mediators of Met's protective effects in GSCs.

Referee #1 (Remarks):

The reviewers have very carefully considered the reviews and have responded very effectively with new data and clarifications. The new data implicating aurora kinase, ATM and p21 strengthen the novel mechanistic aspects of the work. The more expansive in vivo xenograft experiments especially the results showing that Met inhibition sensitizes xenograft GSCs to radiation therapy (depletion of sphere-forming and tumor propagating cells)strengthens previous conclusions.

Referee #2 (Remarks):

The authors have provided a revised version of their manuscript that includes clear answers to my points. I am particularly very happy to see the novel Figure 8 with the serial transplant that supports by itself the all manuscript. I now fully support publication of this work.

Referee #3 (Remarks):

Current version of the manuscript by De Bacco et. al. is a significant improvement over the previous manuscript. With minor edits/changes listed below, manuscript will be in acceptable form.

1. Authors are requested to show same loading control for all western blot experiments. Authors seems to switch between vinculin and b-actin.
2. Authors are also requested to show loading controls for cytoplasmic and nuclear fractions in Figure 7 A-C to validate proper fractionation of protein lysate.

3. Authors are also requested to show full blots for cleaved PARP and Cleaved Caspase (Figure 4B) to clearly show apoptotic activity.

Minor Suggestion: Some details such as concentration of HGF used for supplementation, concentration of different drugs used, time of drug treatment , etc.; if added to figure legends, will make reading experience much better.

2nd Revision - authors' response

26 February 2016

We are extremely pleased and grateful for the very rapid and positive evaluation of our revised manuscript.

We provide a final version with the requested amendments:

- a new Figure 7 showing loading controls for panel C; please note that loading controls for panel A and B were unfeasible, as these are immunoprecipitations on cytoplasmic fractions; however, total p21 was shown as internal control of phosphorylated p21 (Reviewer N. 3);
- revised figure legends, as to indicate concentrations of HGF and inhibitors, and duration of the treatments (Reviewer N. 3)

We also provide as “Source Data” the original, uncropped and unprocessed Western blots shown in Figures 4-7. Please note that some croppings were performed on the original membranes, to decorate the same samples with different antibodies.

Corresponding Author Name: Carla Boccaccio
 Journal Submitted to: EMBO Molecular Medicine
 Manuscript Number: EMM-2015-05890